# Antibacterial Activities of Ag/Cellulose Nanocomposites Derived from Marine Environment Algae against Bacterial Tooth Decay

**DOI:** 10.3390/microorganisms12010001

**Published:** 2023-12-19

**Authors:** Ragaa A. Hamouda, Rabab R. Makharita, Fauzia A. K. Qarabai, Fathi S. Shahabuddin, Amna A. Saddiq, Laila Ahmed Bahammam, Shaymaa W. El-Far, Mamdouh A. Bukhari, Mohammad A. Elaidarous, Asmaa Abdella

**Affiliations:** 1Department of Biology, College of Sciences and Arts at Khulis, University of Jeddah, Jeddah 21959, Saudi Arabia; ragaahom@yahoo.com (R.A.H.); rabab_makharita@science.suez.edu.eg (R.R.M.); foz201414@gmail.com (F.A.K.Q.); aansaddiq@uj.edu.sa (A.A.S.); 2Department of Microbial Biotechnology, Genetic Engineering and Biotechnology Research Institute (GEBRI), University of Sadat City, Sadat City 32897, Egypt; 3Botany and Microbiology Department, Faculty of Science, Suez Canal University, Ismailia 41522, Egypt; 4Harad Center, Makkah Ministry of Health, Riyadh 24342, Saudi Arabia; fat7y201111@gmail.com; 5Department of Endodontics, Faculty of Dentistry, King Abdulaziz University, Jeddah 21589, Saudi Arabia; lbahammam@kau.edu.sa; 6Division of Pharmaceutical Microbiology, Department of Pharmaceutics and Industrial Pharmacy, College of Pharmacy, Taif University, Taif 21974, Saudi Arabia; shfar@tu.edu.sa; 7Regional Laboratory, Laboratories and Blood Banks Administration, Ministry of Health, 7780 Wali Alahed, Makkah P.O. Box 24353-4537, Saudi Arabia; mamdouhb@moh.gov.sa (M.A.B.); melaidarous@moh.gov.sa (M.A.E.); 8Department of Industrial Biotechnology, Genetic Engineering and Biotechnology Research Institute, University of Sadat City, Sadat City 32897, Egypt

**Keywords:** *Ulva lactuca*, Ag/cellulose nanocomposites, fluoride, dental caries, isolation, antimicrobial

## Abstract

Dental caries is an infectious oral disease caused by the presence of different bacteria in biofilms. Multidrug resistance (MDR) is a major challenge of dental caries treatment. Swabs were taken from 65 patients with dental caries in Makkah, Saudi Arabia. Swabs were cultivated on mitis salivarius agar and de Man, Rogosa, and Sharpe (MRS) agar. VITEK 2 was used for the identification of isolated bacteria. Antibiotic susceptibility testing of the isolated bacteria was performed using commercial antibiotic disks. Ulva lactuca was used as a reducing agent and cellulose source to create nanocellulose and Ag/cellulose nanocomposites. Fourier-transform infrared spectroscopy (FTIR), transmission electron microscopy (TEM), scanning electron microscopy (SEM), energy-dispersive X-ray spectroscopy (EDS), and X-ray diffraction spectroscopy (XRD) were used to characterize nanocellulose and Ag/cellulose nanocomposites. The results showed that most bacterial isolates were *Streptococcus* spp., followed by *Staphylococcus* spp. on mitis salivarius media. *Lactobacillus* spp. and *Corynebacterium* group f-1 were the bacterial isolates on de Man, Rogosa, and Sharpe (MRS) media. The antibiotic susceptibility test revealed resistance rates of 77%, 93%, 0, 83%, 79%, and 79% against penicillin G, Augmentin, metronidazole, ampicillin, ciprofloxacin, and cotrimoxazole, respectively. Ag/cellulose nanocomposites and Ag/cellulose nanocomposites with fluoride were the most effective antibacterial agents. The aim of this work was to assess the antibacterial activity of Ag/cellulose nanocomposites with and without fluoride against bacteria isolated from the oral cavities of patients with dental caries. This study demonstrated that Ag/cellulose nanocomposites have antibacterial properties against multidrug-resistant bacteria that cause dental caries.

## 1. Introduction

Nanocelluloses are cellulosic materials with at least one dimension in the nanometer range [1]. There are two types of nanocellulose: cellulose nanocrystals (CNCs) and cellulose nanofibrils (CNFs) [2]. Aside from being renewable, nanocellulose possesses the following properties: stiffness, extreme strength, dimensional stability, low coefficient of thermal expansion, low density, the ability to change its surface chemistry, a high surface-to-volume ratio, and excellent tensile strength [3]. A nanocomposite is formed when nanocellulose and metal nanoparticles are joined. By using cellulose as a soft matrix to hold inorganic fillers such as metal nanoparticles in place, researchers have created composites that combine the fillers’ inherent activities with the biointerfaces provided by cellulose fibers [4]. The polymer acts as a surface-capping, reducing, and/or stabilizing agent when the metal nanoparticles are incorporated into or enclosed inside the polymer matrix [5,6].

The term “oral microbiome” refers to the collective genome of bacteria that inhabit the oral cavity. A core microbiome and a variable microbiome make up the human microbiome. Humans all share the same core microbiome; however, each person has a different variable microbiome because of lifestyle and physiological variations [7]. The microbiota has a significant impact on the promotion of health or the progression of disease [8]. The oral microbiome is typically found in the form of biofilms. It is essential for maintaining oral homeostasis, preserving the oral cavity, and preventing disease. Knowing the identity of the microbiome and its shared neighbors is needed for mechanistic knowledge of the major players [9].

Dental caries, which affects 2.4 billion people worldwide, is one of the most prevalent chronic diseases [10]. Dental caries is a bacterial disease that develops over time as a result of intricate biological interactions between fermentable carbohydrates and acidogenic bacteria [11]. It is marked by demineralization of dental surfaces, which are the result of acidic niches produced by metabolic products of biofilms [12]. *Streptococcus mutans*, a highly acidogenic bacteria that causes dental caries, softens the hard tissues of teeth by producing short-chain acids [13]. Acid creates an acidic environment around the tooth. As a result, the enamel and dentin later become demineralized [14]. Untreated dental caries result in severe consequences, including the risk of dental sepsis [15]. Topical fluoride administration and fissure sealant application are the two basic clinical preventative techniques for minimizing the incidence of tooth cavities. On smooth dental surfaces, topical fluoride treatment is more effective, whereas pit and fissure sealants are employed successfully on the occlusal tooth surface [16]. Established treatments for dental caries include the use of fluorides and control of plaque through professional dental treatments and mouthwashes containing chlorhexidine gluconate [17].

Due to the widespread use of common antibiotics in dental caries treatment, antibiotic resistance in bacteria has developed, endangering public health. [18]. In many circumstances, researchers have focused on new alternative therapies or approaches that can overcome antibiotic resistance while still being cost-effective, such as nanoparticles (NPs) and nanocomposites [19] The antibacterial activity of pure cellulose can only be emphasized in its derivatives, which are created through diverse chemical functionalization with antibacterial groups or by combination with natural or manufactured bioactive components or polymers, as well as metal nanoparticles and metal oxides [20,21]. Because of their distinctive physical and chemical properties, Ag nanoparticles (AgNPs) are widely used in a variety of fields. Because of their large specific surface area and high surface activity, they have great adsorption and are efficient catalysts in heterogeneous catalytic processes [22]. AgNPs have antimicrobial activity due to their combined effects of binding Ag ions to cell walls, inactivation of membrane-associated enzymes, accumulation within cells, interference with essential biomolecules of bacterial cells, denaturation of the cell envelope, and formation of reactive oxygen species. All of these events are critical for bacterial cells [23].

Nanocomposites such as ZnO–CNC have been proven to have antibacterial activities more than ZnO nanoparticles against *S. aureus* and *E. coli.* [24,25]. Ag–Fe3O4–cellulose nanocrystal (CNC) nanocomposites were produced synthetically and showed improved antibacterial activity compared to AgNPs [26]. A constant and localized antibacterial impact is produced through the regulated release of silver ions from these nanocomposites. The method reduces the risk of injury to host cells while also lowering the overall dosage needed [27]. There are several distinct techniques by which silver nanoparticles (AgNPs) can be synthesized. They can be produced using physical, chemical, and biological methods. The biological method is preferred over the other methods because it has less toxicity, high stability, and better physiochemical characteristics [28]. Microorganisms such as bacteria, fungi, algae, and yeast are employed in the green synthesis of AgNPs [29]. Currently, the use of algae for the biosynthesis of nanoparticles is prevalent. The use of algae is mainly due to their high capacity to take in metals and reduce metal ions, relatively low production costs, and most importantly, their ability to produce nanoparticles on a large scale [30]. Nanocellulose production has also been exploited by green algae such as *Ulva*, *Cladophora*, *Chaetomorpha*, *Rhizoclonium*, and *Siphonocladales* [31].

The goal of this study is to synthesize and characterize Ag/cellulose nanocomposites utilizing algae (*Ulva lactuca*). Additionally, it aims to study the antibacterial effects of AgNPs, nanocellulose, and Ag/cellulose nanocomposites with and without fluoride on various bacteria isolated from dental caries patients.

## 2. Materials and Methods

### 2.1. Materials and Chemicals

The materials used in the study were sodium hydroxide (NaOH), ethanol (99%), hydrochloric acid (37%), hydrogen peroxide (6%), silver nitrate 99.9+% (metal basis), and distilled water. All the chemicals used in this research were of analytical grade and applied without further purification. Chemical materials were purchased from the Saudi Chemical Company (PanReac AppliChem, Ar Riyad, Saudi Arabia). Hama fluoride topical gel (1.23% fluoride ion) was obtained from Kal-AlHamaya, Saudi Arabia, antibiotic susceptibility disks (penicillin G (PG), Augmentin (AUG), metronidazole (MZ), ampicillin (AP), ciprofloxacin (CIP), and cotrimoxazole (TS)) from PanReac AppliChem, Riyadh, Saudi Arabia, and Gram-positive cocci ID cards (ID-GPC cards; BioMérieux, Craponne, France.

### 2.2. Algae Collection

*Ulva lactuca* green algae were collected from the Red Sea shore in Jeddah, Saudi Arabia. The alga was identified by Professor Ragaa Hamouda (Professor of Microbiology, University of Sadat City, Egypt). Algae were washed in water to eliminate contaminants before being dried in a 60 °C oven until reaching a constant weight, then milled and sieved using 60-mesh sieves. Fine particles with an average size of 0.3 mm were selected and preserved in a dry place for future use.

### 2.3. Extracting Cellulose from U. lactuca Green Alga

Fifty grams of pulverized alga was poured in a flask with 170 mL of pure ethanol and 30 mL of water and shaken at 60 °C for 6 h before filtering with Whatman^®^ qualitative filter paper, Grade 1 (Merck, Rahway, NJ, USA). The liquid phase was then discarded. The insoluble fraction was washed numerous times with 99% ethanol before being oven-dried for 16 h at 37 °C. After drying, the sample was suspended in 400 mL of 4% H2O2 and heated to 80 °C for 16 h to remove any remaining green pigments and other colored impurities. After cooling to room temperature, the mixture was filtered, and the liquid phase was discarded. The insoluble fraction was then suspended in 400 mL of 0.5 M NaOH after being rinsed with distilled water. For 16 h, the mixture was maintained at 60 °C and was then removed from the oven, allowed to cool to room temperature, filtered, and washed three times with distilled water before the insoluble fraction was collected [32]. The extraction procedure is schematically shown in Figure 1.

### 2.4. Synthesis of Nanocellulose

About 40 g of insoluble fraction was treated for 10 min at 90 °C with 27 mL of 38% HCl and 173 mL of distilled water. After cooling to room temperature, the mixture was centrifuged at 5000 rpm for 15 min, and the insoluble fraction was washed with distilled water. When the mixture was completely dry, it was kept at 60 °C.

### 2.5. Silver Nanoparticle Preparation

Dried *Ulva lactuca* alga (1 g) was extracted by boiling in 100 mL of distilled water for one hour, followed by filtration. Silver nitrate (0.17 g) was added to 90 mL of distilled water. Silver nitrate solution was added dropwise to 10 mL of the prepared algae extract at 60 °C while the mixture was constantly stirred until it turned brown. The mixture was centrifuged at 8000 rpm at 30 °C for 30 min. The supernatant was discarded. The silver nanoparticle pellets were washed 3 times to remove algal residue and were oven-dried at 50 °C for 3 h.

### 2.6. Biosynthesis of Ag/Cellulose Nanocomposites

Silver nitrate (0.085 g) was added to 45 mL of distilled water, Then, 0.4 g nanocellulose was added, and then alga extract (5 mL) added dropwise. The mixture was mixed gently and heated at 60 °C until the color turned brown. The mixture was centrifuged at 8000 rpm at 30 °C for 30 min. The supernatant was discarded. The Ag/cellulose nanocomposites were washed 3 times after drying to remove algal residue and oven-dried at 50 °C for 3 h.

### 2.7. Mixing of Fluoride with Ag/Cellulose Nanocomposites

One milliliter of 1.7 mg/mL silver nanoparticles, 4 mg/mL nanocellulose, and 2 mg/mL Ag/cellulose nanocomposites were prepared and mixed with 10 mL of fluoride topical gel (1.23% fluoride ion) by a magnetic stirrer for 10 min.

### 2.8. Characterization of Ag/Cellulose Nanocomposites

#### 2.8.1. FTIR Spectroscopy Analysis

FTIR was used to determine the functional groups of Ag–cellulose nanocomposites. To obtain pellets for FTIR analysis, the materials were lyophilized and combined with KBr powder. The FTIR spectra were acquired (Waltham, MA, USA) at a resolution of 4 cm^−1^ in the 4000–400 cm^−1^ region.

#### 2.8.2. X-ray Diffraction (XRD)

An X-ray diffractometer (PAN Analytical X-Pert PRO, spectris plc, Almelo, the Netherlands) was used to analyze the X-ray diffraction patterns of nanocellulose and cellulose/silver nanocomposites. Scherrer’s equation was used to calculate the cellulose size:Crystal Size L = λk/β Cos θ
where λ = 0.1540 nm, k is the constant factor of 0.91, θ = diffraction angle in radians, and β = full width at half maximum (FWHM).

#### 2.8.3. Energy-Dispersive Spectroscopy (EDS)

The surface morphology of Ag/cellulose nanocomposites was examined using an energy-dispersive spectroscopy (EDS)-equipped field-emission scanning electron microscope (JEOL JSM-6510/v, Tokyo, Japan).

#### 2.8.4. Scanning Electron Microscopy (SEM)

Using a scanning electron microscope (SEM, JEOL JSM-6510/v, Tokyo, Japan) working at 30 kV, the morphologies of Ag/cellulose nanocomposites were investigated.

#### 2.8.5. Transmission Electron Microscopy (TEM)

TEM (JEOL JSM-6510/v, Tokyo, Japan) was used to study the morphology of cellulose–silver nanocomposites at the nanoscale.

### 2.9. Antibacterial Activity of Ag/Cellulose Nanocomposites/Fluoride against Bacteria Isolated from Dental Caries

#### 2.9.1. Ethical Approval

According to the Helsinki Declaration and its amendments, the study was carried out at the Ministry of Health’s Haradh Health Centre in Makkah, Saudi Arabia, from June to October 2022. The Security Forces Hospital Makkah (SFHM)’s local Human Research Ethics Committee examined and granted approval for the investigations involving human subjects (Reference No. H-02-K-076-0522-731). Written informed consent was provided by the participants in that study.

#### 2.9.2. Collection and Isolation of Bacterial Strains

Dentists at the Haradh Health Center of the Ministry of Health, Makkah, Saudi Arabia conducted face-to-face interviews with 65 adult patients suffering from dental carries. Sterile cotton swabs were taken and dipped in transport media (phosphate-buffered saline). The swabs were then squeezed onto the wall of the clean, dry, sterile test tube and gently pressed against the cavity portion of the tooth or socket cavity and rotated gently 2–3 times and then placed in 5 mL of phosphate-buffered saline (1%). Finally, the tubes were labeled with the specific date and symbols. The samples were kept in a clean, sterile tube and stored in an icebox until they reached the side laboratory of microbiology at the regional laboratory in holy Makkah for processing. Fifty-microliter samples were inoculated into mitis salivarius Agar (M-S) agar and de Man, Rogosa, and Sharpe (MRS) agar plates and then incubated at 37 °C for 24–48 h [33]. M-S agar is a medium used with supplements for the selective isolation of Streptococcus viridans, such as Streptococcus mitis and Streptococcus salivarius, and Enterococcus faecalis, from specimens containing mixed microbial flora [34]. MRS Agar (MRS) is an enriched selective medium for the isolation and cultivation of Lactobacillus found in clinical, dairy, and food specimens [35].

#### 2.9.3. Identification of Bacterial Isolates

The VITEK 2 is an automated microbiology system utilizing growth-based technology. The system is available in three formats (VITEK 2 compact, VITEK 2, and VITEK 2 XL) that differ in increasing levels of capacity and automation. The fully automated VITEK 2 system can provide identification results for Gram-positive cocci in a few hours thanks to the improved sensitivity of its fluorescence-based technology, and this feature represents a major improvement over earlier versions of the same system. Previous research has demonstrated that with pure bacterial cultures, this technique may provide reliable identification and susceptibility data [36]. Morphologically distinct colonies were picked, isolates were purified by streaking on nutrient agar, pure cultures were maintained at 4 °C, and different bacterial isolates were carried on a sterile applicator stick. The microbes was suspended in 3.0 mL of sterile saline (aqueous 0.45% to 0.50% NaCl, pH 4.5 to 7.0) in a 12 × 75 mm transparent plastic test tube. After that, identification cards were placed in the test tubes, and these tubes were put into a special rack (cassette). A vacuum chamber station receives the loaded cassette. Before being loaded into the carousel incubator, inoculated cards are passed using a method that seals them after cutting off the transfer tube. The computer linked to the device was used to enter the data of the entered samples [37].

#### 2.9.4. Antimicrobial Susceptibility Tests

The bacterial isolates’ susceptibility to antibiotics was determined by the Kirby–Bauer disk diffusion method using Müller–Hinton agar [38]. Breakpoints for antibiotics were interpreted depending on the guidelines of CLSI, 2020 [39].

#### 2.9.5. Antibacterial Activities of Cellulose, AgNPs, Nanocellulose, and Ag/Cellulose Nanocomposites with and without Fluoride against Isolated Bacteria

The antibacterial characteristics of hybrids constructed of AgNPs, nanocellulose, and Ag/cellulose nanocomposites, both blended with fluoride, were studied using the agar-well diffusion technique against isolated bacteria A Petri dish was filled with Muller–Hinton agar, which solidified. The turbidity of an isolated bacteria overnight broth culture was adjusted to 0.5 McFarland standards. The plates were then individually spread with 50 µL of bacterial suspension using a sterilized cotton swab. Next, 100 µL of the tested material at concentration of 0.4 mg/mL was placed in wells on each bacterial agar plate. The plates were incubated at 37 °C for 24 h. The inhibitory zone was measured (mm). The experiment was carried out in triplicate [40].

## 3. Results and Discussion

### 3.1. Characterization of Ag/Cellulose Nanocomposites

#### FTIR Spectroscopy Analysis

Figure 2 demonstrates the results of the FT-IR spectroscopy analysis of Ag/cellulose nanocomposites biosynthesized from *Ulva lactuca*. The results display 11 absorption bands with Ag/cellulose nanocomposites. The position of the vibration band at 3419 cm^−1^ is due to the symmetric O-H group [41]. Hydroxyl groups exist on the polysaccharides and monosaccharides of the algal material, which might be included in the production of silver nanoparticles [42]. The position of the vibration band at 2928 cm^−1^ is due to the C-H stretching group [43]. Furthermore, the bending vibration modes of amide I and amide II are situated in the region of 1639 and 1547 cm^−1^, respectively. The amide linkage is most likely involved in the reduction of silver ions to nanoparticles and the conservation of silver nanoparticles in the medium [44,45]. Extract of green algae is rich in amide carboxylic acid and nitro compounds which were used for the synthesis of spherical AgNPs by catalyzing the reduction of silver ions (Ag^+^) to Ag^0^ [46,47]. There is a CH bending vibration in the region of 1384 cm^−1^ [48], and the C-O group was observed at 1234 cm^−1^ [49]. The peak at 1158 cm^−1^ is due to the C-C/C-N stretching group [50]. C=O stretch group appears at 1063 cm ^−1^ [26]. The C-H bending group is present at 669 cm^−1^ [51]. The peak at 602 cm^−1^ was related to C≡C-H group [30].

### 3.2. X-ray Diffraction (XRD) of Ag/Cellulose Nanocomposites

Crystal size and crystalline form have an impact on the peak intensity and shape in XRD patterns [52]. The peaks of XRD diffraction patterns of Ulva–Ag/cellulose nanocomposites recorded at 2θ were 11.51, 20.67, 23.26, 27.71 28.9, 29.56, 30.95, 31.62, 32.13, 36.23, 37.23, 43.37, 45.4, 47.58, 49.06, 54.68, 55.84, 56.49, 57.31, 58.21, 66.09, and 67.33° correspond to lattice plane (hkl) 100, 111, 200, 211, 211, 211, 211, 220, 220, 310, 310, 321, 321, 400, 410, 421, 421, 421, 422, 430, 521, and 521. The peaks located at 2 θ = 11.51, 20.67, 23.26, 27.71 28.9, 29.56,30.95, 31.62, 32.13, are attributed to cellulose. The diffraction peaks at 2θ = 36.23, 37.23, 43.37, 45.4, 47.58, 49.06, 54.68, 55.84, 56.49, 57.31, 58.21, 66.09, and 67.33. were attributed to silver. The main crystalline peak was obtained at 2 θ (32.13°) with an intensity of 100% and crystalline size of 27.70 nm, and all peaks were sharp, and all crystalline sizes were in the nanometer range, which confirmed the crystalline Ulva–Ag/cellulose nanocomposites (Figure 3 and Table 1). The sharpness of the diffraction peaks indicates the high crystallization performance of silver [53].

### 3.3. Energy-Dispersive Spectroscopy (EDS) of Ag/Cellulose Nanocomposites

EDS was used to identify the chemical elements and quantify their relative abundance [53]. Ag/cellulose nanocomposites’ EDS examination revealed that nine components were present—C, O, Na, Mg, Si, Cl, Ca, Fe, Cu, Zn, and Ag—with percentage weights of 33.48, 32.02, 3.01, 0.53, 0.55, 12.52, 6.55, 0.79, 1.42, 1.12, and 8.03, respectively, as shown in Figure 4. The trace elements that appear may be absorbed by algae and not dissolved during the nanocellulose preparations. The study of silver nanoparticles using energy-dispersive spectroscopy (EDS) highlighted how well they were dispersed on the surface of cellulose and entered the cellulose network [54].

### 3.4. Scanning Electron Microscopy (SEM) of Ag/Cellulose Nanocomposites

The SEM image of Ag/cellulose nanocomposites reveals surfaces with valleys and ridges, confirming their well-organized bilayer porosity architecture and substantial surface area (Figure 5A). Tan et al. [55] validated these findings, reporting that the addition of Ag enhanced the number of nonhomogeneous Turing structures.

### 3.5. Transmission Electron Microscopy (TEM) of Ag/Cellulose Nanocomposites

Figure 5B shows TEM images of Ag/cellulose nanocomposites biosynthesized from *Ulva lactuca.* TEM images of Ag/cellulose nanocomposites exhibited polydispersed hexagonal nanoparticles with diameters ranging from 12.37 to 19.12 nm. The tiny particle size results in a large surface area, which increases the antibacterial activity of the nanoparticles [56]. The image shows a shell of nanocellulose surrounding a dark core of AgNPs. One essential way to guarantee the stability of metal nanoparticles is to cap them with a polysaccharide [57]. According to the TEM images, Ag/cellulose nanocomposites show a very good spatial dispersion of AgNPs. The TEM results of AgNP-based cellulose nanocomposites revealed that they are spherical with a size of 10–60 nm.


*Leuconosticmesenteroides*


### 3.6. Isolation and Identification of Bacterial Isolates on Mitis Salivarius (M-S) Agar

As detailed in Table 2, 11 types of bacterial isolates with different percentages were isolated on mitis salivarius (M-S) agar media. The most abundant bacterial isolates were *Streptococcus* spp., followed by *Staphylococcus* spp., while *Kytococcus* spp., *Granulicatella* spp., *Gemella* spp., *Kocuria* spp., *Aero coccus* spp., and *Leuconostoc* spp. had the lowest percentages.

#### Isolation and Identification of Bacterial Isolates on De Man, Rogosa and Sharpe (MRS) Media

Table 3 shows that the bacterial isolates from De Man, Rogosa and Sharpe (MRS) media were *Lactobacillus acidophilus, Lactobacillus plantarum* and *Corynebacterium* group f-1.

Our findings concur with those of other studies that reported that these strains are bacteria isolated from tooth decay. *Streptococcus oralis* and *Streptococcus sanguis* are early colonizers that are converted to substrates for attachment of later colonizers such as *Streptococcus mutans* [58]. Following decades of research and examination of caries locations in both people and animals, *Streptococcus mutans* has been identified as a participant in the dental caries process [59]. *Streptococcus mutans* is a productive caries-associated bacterium due to a variety of virulence characteristics [60]. *Streptococcus mutans* can withstand environmental challenges, especially low pH, encourage bacterial adhesion, and produce many organic acids [61]. According to Alash et al. [62], one-third of swab samples from infected teeth were determined to be *Staphylococcus lentus.* A high percentage of dental samples showed *Staphylococcus hominis* spp. [63]. Lactobacilli have been considered major contributors to human dental caries for over a century [64]. According to Schoilewa et al. [65], *Corynebacterium* was the genus most associated with plaque, and it established a habitat for other species and created a consortium with plaque-characteristic traits.

### 3.7. Antibiotic Susceptibility Pattern for Different Isolates

It is evident from Table 4 that all strains isolated were resistant to metronidazole, while the pattern of multidrug resistance varied among the different antibiotics.

*Streptococcus salivarius* strains isolated from patients 12 and 14 and *Kytococcu* strains isolated from patient 52 were resistant to all antibiotics used. Additionally, *Kocuria kristinae* strains isolated from patients 10, 62, 39, and 55 were sensitive to Augmentin and resistant to penicillin, ampicillin, ciprofloxacin, and cotrimoxazole. *Streptococcus thoraltens* strains isolated from patients 27 and 20 were sensitive to penicillin, Augmentin, ampicillin, ciprofloxacin, and cotrimoxazole. *Streptococcus alactolyticus* strains isolated from patient number 58 were sensitive to penicillin, Augmentin, ampicillin, ciprofloxacin, and intermediate to cotrimoxazole, while isolated strains from patient 48 were sensitive to penicillin, Augmentin, ampicillin, ciprofloxacin, and cotrimoxazole. *Streptococcus sanguinis* strains isolated from patients 47, 30 and 38 were sensitive to Augmentin, ampicillin, ciprofloxacin, and cotrimoxazole, but they were resistant to penicillin and metronidazole.

*Staphylococcus lentus* strains isolated from patients 63, 56, 54, 49, 32 and 28 were sensitive to penicillin, Augmentin, ampicillin, ciprofloxacin and cotrimoxazole. A *Staphylococcus hominis* strain isolated from patient 36 was sensitive to penicillin, Augmentin, ampicillin, ciprofloxacin, and cotrimoxazole.

*Granulicatella adiacens* strains isolated from patients 40 and 43 were sensitive to penicillin, Augmentin, ampicillin, ciprofloxacin, and cotrimoxazole. The *Granulicatella elegans* strain isolated from patient 25 was sensitive to penicillin, Augmentin, ampicillin, ciprofloxacin, and cotrimoxazole. *Gemella morbillorum* strain isolated from patient 60 was sensitive to penicillin, Augmentin, ampicillin, ciprofloxacin and cotrimoxazole.

*Gemella sanguinis* strains were isolated from patients 65 and 51. The first isolate was sensitive to penicillin, Augmentin, ampicillin, and cotrimoxazole, but it was intermediate to ciprofloxacin. The second isolated strain was sensitive to penicillin, Augmentin, ampicillin, ciprofloxacin, and cotrimoxazole. An *Aerococcus viridans* strain isolated from patient 50 was sensitive to penicillin, Augmentin, ampicillin, ciprofloxacin and cotrimoxazole. *Leuconostoc mesenteroides* isolated from patient 61 was sensitive to penicillin, Augmentin, ampicillin, ciprofloxacin, and cotrimoxazole.

*Lactobacillus acidophilus* strains isolated from patients 19, 64, 29, 31, 33, 34, 35, 37, 42, 45, 46, 5, 53, 57, and 59 were sensitive to penicillin, Augmentin, ampicillin, ciprofloxacin and cotrimoxazole. A *Lactobacillus plantarum* strain isolated from patient 9 was resistant to penicillin, ciprofloxacin, and metronidazole, but sensitive to Augmentin, ampicillin and cotrimoxazole. A *Corynebacterium* group f-1 strain isolated from patient 22 was sensitive to penicillin, Augmentin, ampicillin, ciprofloxacin and cotrimoxazole.

The incorrect utilization of antibiotics leads to the evolution of multidrug resistance (MDR) in microbial flora [66]. The World Health Organization has recognized MDR bacteria as one of the top three dangers to human health. Antibiotic resistance occurs when patients stop their medication by themselves or take inadequate doses [67]. The proteins that participate in this process are called MDR proteins [68]. Bacteria have various methods that counteract antibiotics’ effects, including prohibiting antibiotics from entering the bacterial cell [69]. Cao et al. [70] explained the antibiotic resistance of bacteria in biofilms by preventing biofilm-to-antibiotic permeation. The most important mechanism of resistance to penicillins and cephalosporins is antibiotic hydrolysis mediated by the bacterial enzyme β-lactamase [71]. Aminoglycoside antibiotic modification using aminoglycoside modification enzymes (AMEs), including N-acetyl transferases (AAC), O-phosphotransferases (APH), and O-adenyltransferases (ANT), produced by bacteria is a commonly used strategy for rendering an antibiotic ineffective [72]. Metronidazole resistance can occur by several mechanisms that involve reduced uptake of the drug, increased removal from the bacterial cell or by reducing the rate of metronidazole activation inside anaerobes [73]. According to Reygeart et al., [74] the basic mechanisms of resistance are drug uptake limitation, drug target alteration, drug inactivation, and active efflux of a drug.

### 3.8. Antibacterial Activity of Cellulose, Nanocellulose, AgNPs, and Ag/Cellulose Nanocomposites with and without Fluoride against Antibiotic-Resistant Bacterial Isolates

Cellulose–fluoride, nanocellulose/fluoride, AgNPs, AgNPs/fluoride, Ag/cellulose, and Ag/cellulose nanocomposites with fluoride possessed antibacterial activity against isolated bacteria. Ag/cellulose nanocomposite with fluoride was the most effective antibacterial agent. Generally, nanobased formulation is a promising strategy for improving clinical application. Interestingly, Figure 6 shows the efficacy of the ulose nanocomposite with fluoride as an antimicrobial agent against *Streptococcus salivarius* isolated from patient 12.

According to the findings in Table 5, 0.4 mg/mL (fluoride, cellulose, or nanocellulose) alone have no antibacterial activity. Nanocellulose blended with fluoride was effective against isolated strains from patients 19, 10, 51, 43, 40, 28, 32, 63, 38, 30, 47, 40, 14, 12, 11, 8, and 48, while cellulose blended with fluoride was effective against isolated strains from patients 20, 48, 8, 40, 32, 25, and 10. AgNPs, AgNPs/fluoride, nanocellulose/AgNPs, and nanocellulose–AgNPs with fluoride at a concentration of 0.4 mg/mL had antibacterial activity against all isolated strains.

According to Coma et al. [21], the antibacterial activity of pure cellulose can only be highlighted in its derivatives, which are obtained through various chemical functionalization with antibacterial groups or by combination with natural or synthetic bioactive components or polymers, as well as metal nanoparticles and metal oxides. The addition of fluoride to cellulose or nanocellulose is necessary for antibacterial activities. Fluoride prevents decay through three mechanisms: inhibition of bacterial enzymes, reinforcement of remineralization at crystal surfaces, and prevention of purification of crystals [75]. Mitwalli et al. [76] developed a novel dental nanocomposite containing nanoparticles of calcium fluoride (nCaF2) for preventing recurrent caries via antibacterial, protein-repellent and fluoride-releasing capabilities.

AgNPs could be an alternative to antibiotics to dominate microbial infections caused by MDR bacteria due to their significantly higher surface area, which results in greater contact with bacteria. These nanoparticles work in several mechanisms, including disrupting bacterial cell membranes by converting Ag^0^ to Ag^+^ and by obstructing intercellular metabolic pathways after Ag^+^ has entered the cell [77]. Liao et al. [78] showed that the MIC and MBC of AgNPs against drug-resistant and multidrug-resistant *P. aeruginosa* were 1.406–5.625 µg/mL and 2.813–5.625 µg/mL, respectively. The primary mechanism is the imbalance in oxidation and antioxidation processes, as well as the failure to remove excess ROS. Bacterial resistance strategies include the acquisition of resistance genes from other bacteria, the formation of biofilms, obstacles to antibiotic permeation, the alteration of the antibiotic target, and efflux pump systems that remove the antibiotic from intracellular media [79]. However, AgNPs act on bacterial cells via a number of concurrent mechanisms, including the diffusion of small AgNPs and Ag ions into the cell, cell membrane rupture and mechanical damage, and protein and DNA alteration in the intracellular medium [80].

Ag/cellulose nanocomposites showed good antibacterial and biofilm formation efficiency against Gram-positive and Gram-negative bacteria [81]. Because nanocellulose fibrils are biologically compatible, they enable interactions between nanocomposites and microorganisms. As a result, nanocellulose templates are thought to improve the antibacterial activity of nanocomposites [82]. Because of its abundance of hydroxyl groups, nanocellulose can adsorb metallic cations. Because of their strong adsorption of Ag^0^ and Ag^+^, the OH groups result in high-density immobilization of AgNPs on nanocellulose [83]. The inclusion of silver nanoparticles is facilitated by the enormous surface area of nanocellulose. The chemical binding of silver to the cellulose matrix results in the formation of the cellulose/Ag complex. As a result, the cellulose-/Ag combination has antibacterial action [84].

## 4. Conclusions

Nanocomposites have resulted in a rapid increase in applications in a variety of disciplines. The effect of nanocellulose and Ag/cellulose nanocomposites biosynthesized from the marine alga *Ulva lactuca* on bacteria isolated from individuals with dental caries who are resistant to known antibiotics was studied in this work. Data analysis demonstrated that Ag/cellulose nanocomposites and Ag/cellulose nanocomposites combined with fluoride have potent antibacterial activity. Based on the data gathered, it is possible to conclude that addition of fluoride or Ag^+^ to nanocellulose is necessary for antibacterial activities. Ag/cellulose nanocomposites have the potential to be an affordable and safe nanocomposite product from a natural source with antibacterial components against multidrug-resistant bacteria. These components could be added to toothpaste and used as tooth fillers. The application of Ag/cellulose nanocomposites as coating agents for dental implants can be used to significantly improve patients’ oral health, leading to their widespread adoption.

## Figures and Tables

**Figure 1 microorganisms-12-00001-f001:**
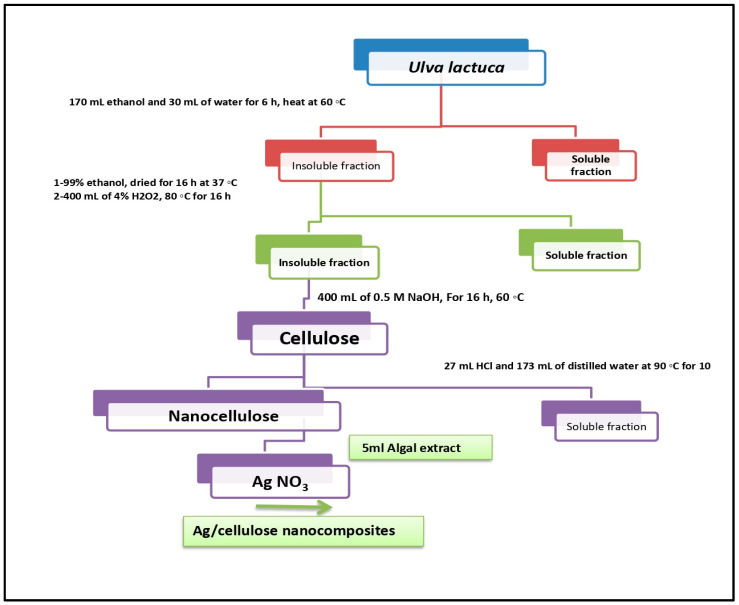
Protocol of cellulose extraction, nanocellulose, and Ag–cellulose nanocomposite synthesis (Reprinted/adapted with permission from Hamouda et al., [32]).

**Figure 2 microorganisms-12-00001-f002:**
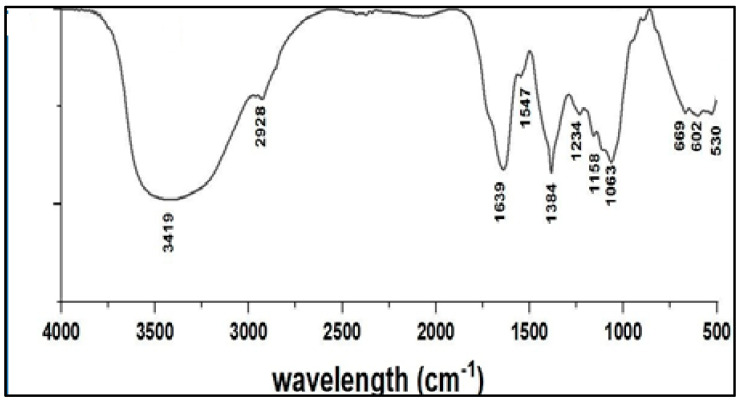
FT-IR spectroscopy of Ag/cellulose nanocomposites.

**Figure 3 microorganisms-12-00001-f003:**
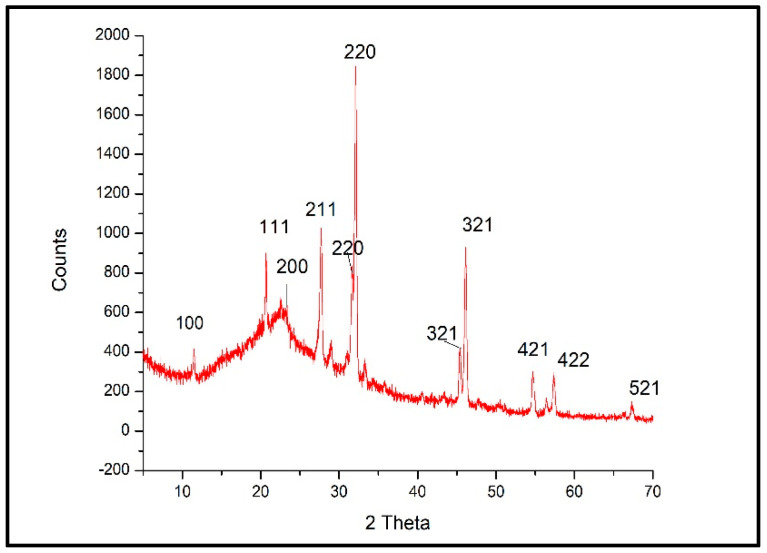
XRD analysis of Ulva/Ag/cellulose nanocomposites derived from *U. lactuca*.

**Figure 4 microorganisms-12-00001-f004:**
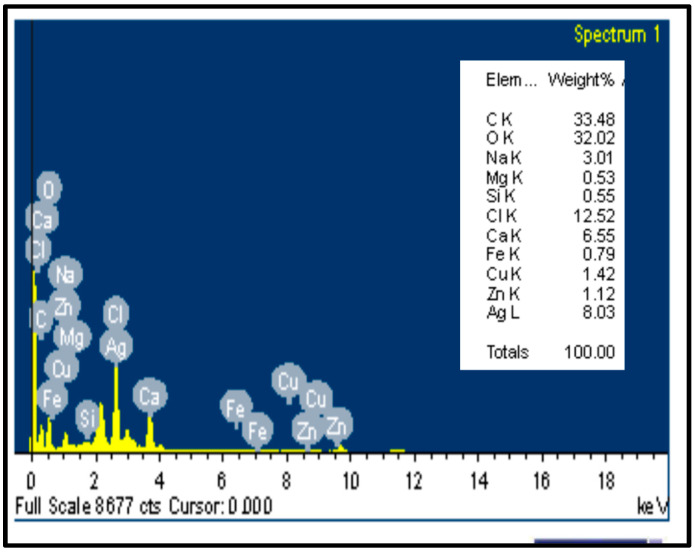
Energy dispersive X-ray spectroscopy (EDS) of Ag–cellulose nanocomposites.

**Figure 5 microorganisms-12-00001-f005:**
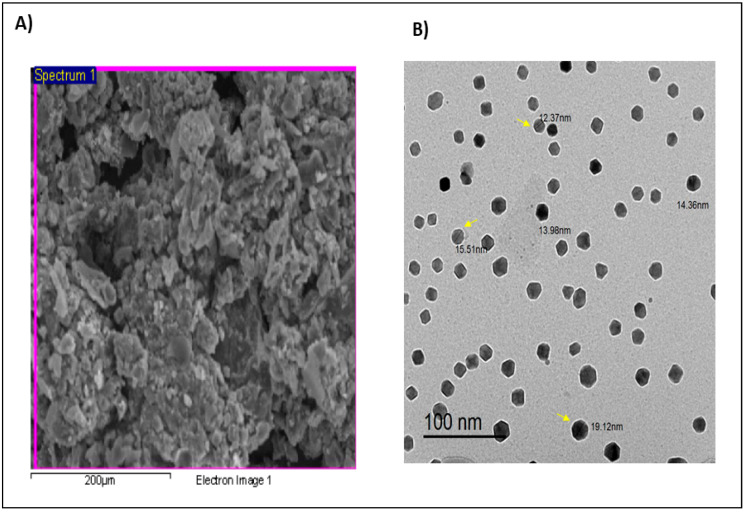
Electron microscopy imaging of ulose nanocomposites. (**A**) Scanning electron microscopy (SEM) image; (**B**) transmission electron microscopy (TEM) image.

**Figure 6 microorganisms-12-00001-f006:**
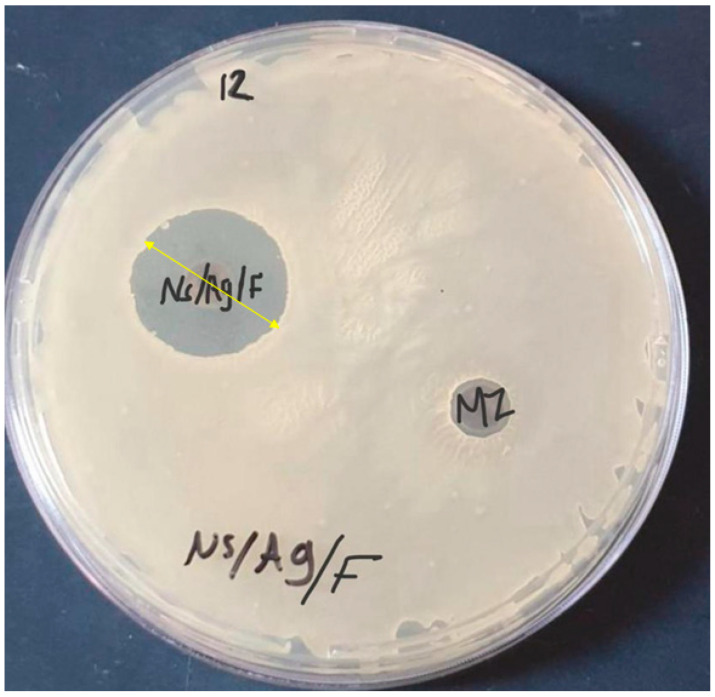
Antimicrobial susceptibility testing of Ag/cellulose nanocomposite with fluoride showed excellent efficacy against *Streptococcus salivarius.* MZ: metronidazole; NS/Ag/F: Ag/Cellulose nanocomposite with fluoride.

**Table 1 microorganisms-12-00001-t001:** Simple peak indexing of Ulva/Ag/cellulose nanocomposites.) derived from *U. lactuca*.

2 Theta	Sin Theta	hkl	D (nm)	Intensity %
11.5169	0.100334799	100	40.16	5.58
20.6775	0.17946759	111	33.08	10.51
23.2611	0.2016002	200	12.35	3.15
27.7198	0.239548602	211	26.12	45.55
28.9014	0.249546871	211	35.53	6.15
29.5627	0.25513104	211	65.04	0.95
30.9599	0.266901148	211	29.93	10.64
31.6265	0.272502759	220	38.83	32.05
32.1345	0.276765431	220	27.70	100
36.2324	0.310945169	310	42.63	2.42
37.2304	0.319210732	310	38.25	2.83
43.3728	0.369526204	321	20.84	1.3
45.4086	0.385975277	321	24.82	20.18
47.5831	0.403410353	400	15.39	1
49.0656	0.415214098	410	61.19	0.87
54.6861	0.4593171	421	21.08	14.53
55.8432	0.468262953	421	35.66	1.01
56.497	0.473296605	421	22.22	4.54
57.3153	0.479575026	422	23.61	13.84
58.2132	0.486436029	430	44.09	0.6
66.0993	0.545365585	521	37.59	1.66
67.3334	0.554360806	521	27.54	5.48

**Table 2 microorganisms-12-00001-t002:** Identification of bacteria isolated on mitis salivarius agar (M-S) using VITEK 2 identification cards.

Test	*Strepto* *thoraltensis*	*Strepto* *alactolyticus*	*Strepto* *mutans*	*Strepto salivarius*	*Strepto* *sanguinis*	*Staph* *lentus*	*Staph* *hominis*	*Granulicatella* *adiacens*	*Gemella sanguinis*	*Kytococcus*	*Kocuria* *kristinae*	*Aerococcus* *viridans*	*Leuconosticme* *senteroides*
**D-amygdalin**	+	+	+	+	+	+	−	−	−	−	**−**	**+**	**−**
**Ala-Phe-Pro arylamidase**	+	+	−	+	−	−	−	−	−	−	**−**	**−**	**−**
**Leucine arylamidase**	+	+	−	+	+	−	−	+	−	−	**−**	**−**	**−**
**Alanine arylamidase**	+	+	+	+	+	−	−	−	−	+	**−**	**−**	**−**
**D-ribose**	−	−	−	−	−	−	−	−	−	−	**−**	**−**	**−**
**Novobiocin resistance**	−	+	−	+	−	−	+	+	−	−	**−**	**−**	**−**
**Pullulan**	+	+	−	−	−	+	−	−	−	−	**−**	**−**	**−**
**Bacitracin resistance**	+	+	−	+	−	−	+	+	−	−	**−**	**−**	**−**
**Alpha-glucosidase**	−	−	−	+	−	−	+	−	−	−	**−**	**−**	**−**
**D-galactose**	+	+	+	+	+	+	−	−	−	−	**+**	**+**	**−**
**Beta-glucuronidase**	−	−	−	−	−	−	−	−	−	−	**−**	**−**	**−**
**Phosphatase**	−	−	−	−	−	−	+	−	−	−	**−**	**−**	**−**
**Arginine dihydrolase 2**	−	−	−	−	−		−	−	−	−	**−**	**−**	**−**
**D-trehalose**	+	+	+	−	+	−	+	−	+	−	**−**	**−**	**−**
**Alpha-mannosidase**	−	−	−	−	−	−	−	−	−	−	**−**	**−**	**−**
**Saccharose/sucrose**	+	+	+	+	+	+	+	+	−	−	**+**	**+**	**+**
**L-lactate alkalinization**	−	−	−	−	−	−	+	−	−	−	**−**	**−**	**−**
**beta-glucuronidase**	−	−	−	−	−	−	−	−	−	−	**−**	**−**	**−**
**N-acetyl-d-glucosamine**	+	+	−	+	+	+	+	−	−	−	**−**	**−**	**−**
**Urease**	−	−	−	+	+	−	+	−	−	−	**−**	**−**	**−**
**D-sorbitol**	+	+	-	−	−	+	−	−	−	−	**−**	**−**	**−**
**Salicin**	+	−	+	+	+	+	−	−	−	−	**+**	**+**	**+**
**arginine dihydrolase 1**	−	−	−	−	−	−	+	+	−	−	**−**	**−**	**−**
**D-xylose**	+	+	−	−	−	+	−	−	−	−	**−**	**+**	**−**
**phosphatidylinositol phospholipase c**	−	−	−	−	−	−	−	−	−	−	**−**	**−**	**−**
**Growth in 6.5% Nacl**	−	−	−	−	−	−	+	−	−	−	**−**	**−**	**−**
**Novobiocin resistance**	−	+	−	+	−	−	+	+	−	−	**−**	**−**	**−**

**Table 3 microorganisms-12-00001-t003:** Identification of bacteria isolated on De Man, Rogosa, and Sharpe (MRS) using VITEK 2 identification cards.

Test	*Lactobacillus acidophilus*	*Lactobacillus plantarum*	*Corynebacterium* Group f-1
**Ala-Phe-Pro arylamidase**	+	−	−
**Beta-galactosidase**	+	−	−
**Alpha-mannosidase**	−	−	−
**Beta-glucuronidase**	−	−	−
**D-galactose**	+	+	+
**L-lactate alkalinization**	−	−	−
**D-Ribose**	−	+	+
**D-sorbitol**	−	+	−
**D-trehalose**	+	+	−
**D-xylose**	−	−	−
**D-sorbitol**	−	+	−
**D-galactose**	+	+	+
**Beta-glucuronidase**	−	−	−
**Alpha-mannosidase**	−	−	−
**Saccharose/sucrose**	+	+	+
**Urease**	+	−	+
**Ellman**	−	−	−
**Succinate alkalinization**	−	−	−
**Phosphatase**	−	−	+
**Alpha Glucosidase**	−	−	+
**Arginine dihydrolase 2**	−	−	−
**Pullulan**	+	+	−
**Bacitracin Resistance**	+	−	+
**D-Maltose**	+	+	+

**Table 4 microorganisms-12-00001-t004:** Antibiotic susceptibility pattern for different isolates.

Strain	Strain Code	Zone of Inhibition (mm)					
		PG (10 µg)	AUG (30 µg)	MZ (5 µg)	AP (10 µg)	CIP (5 µg)	TS (25 µg)
*Streptococcus thoraltensi*	27	30 (S)	30 (S)	0 (R)	30 (S)	30 (S)	30 (S)
	20	30 (S)	33 (S)	0 (R)	30 (S)	30 (S)	25 (S)
*Streptococcus alactolyticus*	58	19 (S)	28 (S)	0 (R)	23 (S)	24 (S)	14 (I)
	48	15 (S)	30 (S)	0 (R)	24 (S)	21 (S)	25 (S)
*Streptococcus mutans*	8	16 (S)	30 (S)	0 (R)	23 (S)	26 (S)	13 (I)
	11	22 (S)	33 (S)	0 (R)	30 (S)	40 (S)	0 (R)
*Streptococcus salivarius*	12	0 (R)	0 (R)	0 (R)	0 (R)	0 (R)	0 (R)
	14	0 (R)	0 (R)	0 (R)	0 (R)	0 (R)	0 (R)
*Streptococcus sanguinis*	47	12 (R)	19 (S)	0 (R)	17 (S)	30 (S)	20 (S)
	30	13 (R)	28 (S)	0 (R)	30 (S)	28 (S)	15 (S)
	38	14 (R)	35 (S)	0 (R)	28 (S)	28 (S)	27 (S)
*Staphylococcus lentus*	63	18 (S)	28 (S)	0 (R)	17 (S)	25 (S)	25 (S)
	56	17 (S)	30 (S)	0 (R)	28 (S)	25 (S)	27 (S)
	54	24 (S)	35 (S)	0 (R)	35 (S)	25 (S)	25 (S)
	49	20 (S)	30 (S)	0 (R)	25 (S)	27 (S)	30 (S)
	32	20 (S)	30 (S)	0 (R)	30 (S)	30 (S)	30 (S)
	28	15 (S)	30 (S)	0 (R)	32 (S)	27 (S)	25 (S)
*Staphylococcus hominis* spp.	36	18 (S)	23 (S)	0 (R)	22 (S)	36 (S)	22 (S)
*Granulicatella adiacens*	40	20 (S)	30 (S)	0 (R)	30 (S)	30 (S)	30 (S)
	43	15 (S)	35 (S)	0 (R)	33 (S)	30 (S)	30 (S)
*Granulicatella elegans*	25	30 (S)	30 (S)	0 (R)	30 (S)	22 (S)	30 (S)
*Gemella morbillorum*	60	25 (S)	30 (S)	0 (R)	35 (S)	24 (S)	25 (S)
*Gemella sanguinis*	65	20 (S)	30 (S)	0 (R)	30 (S)	18 (I)	20 (S)
	51	15 (S)	28 (S)	0 (R)	28 (S)	25 (S)	29 (S)
*Kytococcus*	52	0 (R)	0 (R)	0 (R)	0 (R)	0 (R)	0 (R)
*Kocuria kristinae*	10	0 (R)	25 (S)	0 (R)	0 (R)	0 (R)	0 (R)
	62	0 (R)	20 (S)	0 (R)	0 (R)	0 (R)	0 (R)
	39	12 (R)	30 (S)	0 (R)	0 (R)	0 (R)	0 (R)
	55	13(R)	30(S)	0(R)	0(R)	0(R)	0 (R)
*Aerococcus viridans*	50	20 (S)	32 (S)	0 (R)	30 (S)	21 (S)	25 (S)
*Leuconostoc mesenteroides ssp cremoris*	61	20 (S)	30 (S)	0 (R)	30 (S)	25 (S)	22 (S)
*Lactobacillus acidophilus*	19	25 (S)	30 (S)	0 (R)	30 (S)	30 (S)	30 (S)
	64	20 (S)	20 (S)	0 (R)	28 (S)	26 (S)	32 (S)
	29	25 (S)	30 (S)	0 (R)	30 (S)	25 (S)	25 (S)
	31	15 (S)	26 (S)	0 (R)	25 (S)	25 (S)	22 (S)
	33	16 (S)	27 (S)	0 (R)	23 (S)	25 (S)	20 (S)
	34	15 (S)	25 (S)	0 (R)	25 (S)	25 (S)	20 (S)
	35	17 (S)	20 (S)	0 (R)	19 (S)	25 (S)	25 (S)
	37	15 (S)	24 (S)	0 (R)	24 (S)	25 (S)	24 (S)
	42	25 (S)	30 (S)	0 (R)	25 (S)	25 (S)	25 (S)
	45	23 (S)	35 (S)	0 (R)	26 (S)	25 (S)	25 (S)
	46	21 (S)	30 (S)	0 (R)	28 (S)	23 (S)	26 (S)
	5	20 (S)	35 (S)	0 (R)	17 (S)	25 (S)	25 (S)
	53	17 (S)	33 (S)	0 (R)	23 (S)	23 (S)	25 (S)
	57	19 (S)	30 (S)	0 (R)	25 (S)	20 (I)	22 (S)
	59	16 (S)	30 (S)	0 (R)	27 (S)	22 (S)	23 (S)
*Lactobacillus plantarum*	9	0 (R)	35 (S)	0 (R)	30 (S)	0 (R)	30 (S)
*Corynebacterium* group f-1	22	15 (S)	22 (S)	0 (R)	22 (S)	25 (S)	21 (S)
** *Resistant percentage %* **		23%	7%	100%	17%	21%	21%

Penicillin G (PG), Augmentin (AUG), metronidazole (MZ), ampicillin (AP), Ciprofloxacin (CIP), cotrimoxazole (TS).

**Table 5 microorganisms-12-00001-t005:** Inhibition zone diameter (mm) of different bacterial isolates treated with 0.4 mg/mL of (cellulose, nanocellulose, AgNPs, and Ag/cellulose nanocomposites with and without fluoride).

Strain	Patient Number	Cellulose	F	NC	Cellulose/F	NC/F	AgNPs	AgNPs/F	Ag/NC	Ag/NC/F
*Streptococcus thoraltensi*	27	nd	nd	nd	nd	15 ± 0.28	22 ± 0.17	19 ± 0.11	20 ± 0.57	20 ± 0.57
	20	nd	nd	nd	10 ± 0.2	20 ± 0.0	20 ± 0.0	20 ± 0.23	21 ± 0.3	20 ± 1.15
*Streptococcus alactolyticus*	58	nd	nd	nd	nd	nd	20 ± 0.72	20 ± 0	22 ± 0	23 ± 0.57
	48	nd	nd	nd	17 ± 0.57	20 ± 0.0	20 ± 0.0	13 ± 0	20 ± 0.72	20 ± 0
*Streptococcus mutans*	8	nd	nd	nd	23 ± 0.28	22 ± 0.17	20 ± 0.72	22 ± 0.4	24 ± 0.28	26 ± 1
	11	nd	nd	nd	nd	20 ± 0.35	20 ± 0.11	21 ± 0.15	18 ± 0.41	20 ± 0.57
*Streptococcus salivarius*	12	nd	nd	nd	nd	16 ± 0.28	19 ± 0.0	15 ± 0.28	20 ± 041	22 ± 0.57
	14	nd	nd	nd	nd	15 ± 0.23	18 ± 0.25	19 ± 0.0	14 ± 0.28	20 ± 1.15
*Streptococcus sanguinis*	47	nd	nd	nd	nd	22 ± 0.57	20 ± 0.25	21 ± 0.0	22 ± 0.17	25 ± 0.57
	30	nd	nd	nd	nd	13 ± 0.4	20 ± 0.0	22 ± 0.11	22 ± 0.2	21 ± 0
	38	nd	nd	nd	nd	22 ± 0.28	20 ± 0.0	22 ± 0.17	20 ± 0.4	23 ± 0.57
*Staphylococcus lentus*	63	nd	nd	nd	nd	16 ± 0.0	22 ± 0.41	21 ± 0.45	23 ± 0.23	23 ± 1
	56	nd	nd	nd	nd	nd	21 ± 0.05	22 ± 0.57	20 ± 0.5	20 ± 0.7
	54	nd	nd	nd	nd	nd	20 ± 0.23	21 ± 0.57	20 ± 0	22 ± 1.15
	49	nd	nd	nd	nd	nd	20 ± 0.36	23 ± 0.17	23 ± 0	25 ± 1.15
	32	nd	nd	nd	10 ± 0.0	21 ± 0.28	20 ± 0.35	22 ± 0.28	19 ± 0.23	20 ± 0.57
	28	nd	nd	nd	nd	20 ± 0.2	18 ± 0.75	20 ± 0.25	22 ± 0	21 ± 0.14
*Granulicatella adiacens*	36	nd	nd	nd	nd	nd	23 ± 0.0	22 ± 0.48	22 ± 0.28	25 ± 1.17
*Granulicatella elegans*	40	nd	nd	nd	25 ± 0.28	27 ± 0.57	23 ± 0.57	25 ± 0.46	25 ± 0.1	22 ± 0.3
	43	nd	nd	nd	nd	29 ± 0.0	20 ± 0.28	19 ± 0.17	25 ± 0	27 ± 0.57
*Gemella morbillorum*	25	nd	nd	nd	19 ± 0.17	nd	23 ± 0.23	24 ± 0.11	20 ± 0.11	21 ± 0.8
*Gemella morbillorum*	60	nd	nd	nd	nd	nd	21 ± 0.0	22 ± 0.41	26 ± 0.23	24 ± 0.13
*Gemella sanguinis*	65	nd	nd	nd	nd	nd	2.0 ± 0.57	20 ± 0.57	21 ± 0.4	20 ± 0.57
	51	nd	nd	nd	nd	10 ± 0.28	22 ± 0.57	15 ± 0.42	20 ± 0	20 ± 1.15
*Kytococcus*	52	nd	nd	nd	nd	nd	20 ± 0.46	19 ± 0.2	21 ± 0	23 ± 1.7
*Kocuria kristinae*	10	nd	nd	nd	20 ± 0.14	20 ± 0.17	15 ± 0.57	18 ± 0.25	21 ± 0.28	21 ± 0.57
	62	nd	nd	nd	nd	nd	20 ± 0.36	21 ± 0.057	23 ± 00.17	25 ± 0.57
	39	nd	nd	nd	nd	nd	20 ± 0.23	2.1 ± 0.23	20 ± 0.57	20 ± 1.15
	55	nd	nd	nd	nd	nd	14 ± 0.28	13 ± 0.57	14 ± 0.26	10 ± 0
*Aerococcus viridans*	50	nd	nd	nd	nd	nd	22 ± 0.41	22 ± 0.41	12 ± 0.2	20 ± 0.28
*Leuconostoc mesenteroides*	61	nd	nd	nd	nd	nd	20 ± 0.20	20 ± 0.72	20 ± 0.23	22 ± 0.28
*Lactobacillus acidophilus*	19	nd	nd	nd	nd	nd	20 ± 0.11	22 ± 0.0	20 ± 0.36	22 ± 0.23
	64	nd	nd	nd	0	0	19 ± 0.2	18 ± 0.0	20 ± 0	20 ± 0.11
	29	nd	nd	nd	0	0	20 ± 0.0	20 ± 0.2	20 ± 0	25 ± 0.46
	31	nd	nd	nd	nd	nd	20 ± 0.69	20 ± 0.35	20 ± 0.45	12 ± 0.46
	33	nd	nd	nd	nd	nd	25 ± 0.23	23 ± 0.47	12 ± 0.0	22 ± 0.46
	34	nd	nd	nd	nd	nd	20 ± 0.51	20 ± 0.57	20 ± 0.34	19 ± 0.4
	35	nd	nd	nd	nd	nd	20 ± 0.0	20 ± 0.41	20 ± 0.57	17 ± 0.3
	37	nd	nd	nd	nd	nd	20 ± 0.57	20 ± 0.40	20 ± 0.2	18 ± 0.25
	42	nd	nd	nd	nd	nd	2.0 ± 0.41	19 ± 0.0	22 ± 0.0	25 ± 0.57
	45	nd	nd	nd	nd	nd	12 ± 0.0	22 ± 0.4	25 ± 0.2	27 ± 0.4
	46	nd	nd	nd	nd	nd	24 ± 0.28	21 ± 0.11	22 ± 0.0	23 ± 0.34
	5	nd	nd	nd	nd	nd	20 ± 0.60	20 ± 0.28	22 ± 0.11	23 ± 0.28
	53	nd	nd	nd	nd	nd	23 ± 0.28	23 ± 0.58	25 ± 0.11	25 ± 0.28
	57	nd	nd	nd	nd	nd	20 ± 0.52	21 ± 0	22 ± 0	14 ± 0.82
	59	nd	nd	nd	nd	nd	22 ± 0.0	22 ± 0.28	20 ± 0	25 ± 0.28
*Lactobacillus plantarum*	9	nd	nd	nd	nd	nd	20 ± 0.0	17 ± 0.58	20 ± 0.23	21 ± 0.23
*Corynebacterium* group f-1	22	nd	nd	nd	nd	nd	20 ± 0.57	18 ± 0.25	20 ± 0.23	19 ± 0.57

Fluoride (F), nanocellulose (NC), AgNPs–fluoride (AgNPs–F), Ag–cellulose nanocomposites (Ag–NC), Ag–cellulose nanocomposites with fluoride (Ag/NC/F). nd—not defined.

## Data Availability

Available on request.

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
