# Peer review of "Antibacterial Activities of Ag/Cellulose Nanocomposites Derived from Marine Environment Algae against Bacterial Tooth Decay"

_microorganisms, 2023, doi:10.3390/microorganisms12010001_

Round 1
Reviewer 1 Report (Previous Reviewer 3)
Comments and Suggestions for Authors
The manuscript "Antibacterial activities ofAg/cellulose nanocomposites derived from marine environments algae against bacterial tooth decay" studies antimicrobial properties of nanocomposites Ag/cellulose and Ag/cellulose/F obtained from Ulva Lactuca via against bacteria that cause dental caries.
How did the authors prove that they obtained nanocellulose? There is no such a conclusion after any of the methods in the Discussion part.
The manuscript needs a good proofreading. There are numerous typos, glued words, missed spaces and subscripts.
Title. Please check if the word “activities” is suitable in this particular case. Also, as far as I know, the word “environment” is uncountable.
Line 24. What was conducted on 65 patients?
Lines 100-104. It was already mentioned in the first paragraph of the introduction.
Line 320. The image does not show that it is a shell of NC surrounding a core of AG NP. It only shows some nanoparticles.
Table 2,3. Why some test names in the table and below are written in CAPS?
Line 495. "Ag/cellulose nanocomposites outperformed nanocellulose" NC has never demonstrated antimicrobial activity. How is it possible to outperform something that does not even have this property?
Line 498. "pure cellulose has no antibacterial activities" It is a well-known fact, there was no need to prove obvious things.
Line 500. "Affordable" compared to what? Is it cheaper than antibiotics? Did the authors conduct economic analysis? "safe" Safe to whom/what? How was that proved?
Line 502. Toothpaste contacts with tooth for maximum 4-5 minutes (usually less) and then is removed, and the teeth are thoroughly rinsed with water. Do the authors suppose that for this time Ag/cellulose/F, an additive, will kill dental bacteria?
Author Response
Comments and Suggestions for Authors
The manuscript "Antibacterial activities ofAg/cellulose nanocomposites derived from marine environments
algae against bacterial tooth decay" studies antimicrobial properties of nanocomposites Ag/cellulose and
Ag/cellulose/F obtained from Ulva Lactuca via against bacteria that cause dental caries.
How did the authors prove that they obtained nanocellulose? There is no such a conclusion after any of the methods in the Discussion part.
In both the materials and methods part and the results and discussion part , all data with details
regarding the nano-composite preparation and formulation were mentioned, the nanocomposite was
successfully formulated and completely characterized using FTIR spectroscopy, X-ray diffraction (XRD),
Energy-dispersive spectroscopy (EDS), scanning Electron Microscopy (SEM), Transmission electron
microscopy (TEM)
The manuscript needs a good proofreading. There are numerous typos, glued words, missed spaces and subscripts.
Thanks for your valuable comments
Manuscript was revised.
Title. Please check if the word “activities” is suitable in this particular case. Also, as far as I know, the word “environment” is uncountable.
Thanks for your valuable comments
The word antibacterial activities is commonly used in such cases, environment was used instead of environments.
Here you are some examples of articles used the word antibacterial activity in their titles.
Hamouda RA, Qarabai FAK, Shahabuddin FS, Al-Shaikh TM, Makharita RR. Antibacterial Activity of
Ulva/Nanocellulose and Ulva/Ag/Cellulose Nanocomposites and Both Blended with Fluoride against
Bacteria Causing Dental Decay. Polymers (Basel). 2023 Feb 20;15(4):1047. doi:
10.3390/polym15041047.
Ricardo J. B. Pinto, Sara Daina, Patrizia Sadocco, Carlos Pascoal Neto, Tito Trindade. Antibacterial
Activity of Nanocomposites of Copper and Cellulose. BioMed Research International Volume 2013,
Article ID 280512, 6 pages
Line 24. What was conducted on 65 patients?
Thanks for your valuable comments
Swabs were taken from 65 patients with dental caries in Makkah, Saudi Arabia. Then swabs were cultivated on Mitis Salivarius agar and de Man, Rogosa, and Sharpe (MRS) agar.
It was added to revised manuscript.
Lines 100-104. It was already mentioned in the first paragraph of the introduction.
Thanks for your valuable comments.
Lines were deleted in the revised manuscript.
Line 320. The image does not show that it is a shell of NC surrounding a core of AG NP. It only shows some nanoparticles.Thanks for your valuable comments
Here you are TEM image and the arrow refers to the black particle of Ag and the white line is shell of nanocellulose
Table 2,3. Why some test names in the table and below are written in CAPS?
Thanks for your valuable comments.
They are corrected in the revised manuscript.
Line 495. "Ag/cellulose nanocomposites outperformed nanocellulose" NC has never demonstrated antimicrobial activity. How is it possible to outperform something that does not even have this property?
Thanks for your valuable comments
Sentence was rewritten in the revised manuscript
Line 498. "pure cellulose has no antibacterial activities" It is a well-known fact, there was no need to prove obvious things.
Thanks for your valuable comments
Sentence was deleted in the revised manuscript
Line 500. "Affordable" compared to what? Is it cheaper than antibiotics? Did the authors conduct economic analysis? "safe" Safe to whom/what? How was that proved?
Thanks for your valuable comments
Cellulose was extracted from Ulva lactuca green algae which were collected from sea. From Energy dispersive spectroscopy, it is shown that weight % of Ag is only 8.08%. Also, the incidence of microbial
resistance remains low compared to antibiotic, so Affordable expression is a general word that expresses how Ag/cellulose nano-composites as whole are reasonable compared to antibiotics.
This concept is well known according to previously published articles,
Chopra, I. The increasing use of silver-based products as antimicrobial agents: a useful development or a
cause for concern? Journal of Antimicrobial Chemotherapy, Volume 59, Issue 4, April 2007, Pages 587–
590, https://doi.org/10.1093/jac/dkm006
Fiorati, A.; Bellingeri, A.; Punta, C.; Corsi, I.; Venditti, I. Silver Nanoparticles for Water Pollution Monitoring
and Treatments: Ecosafety Challenge and Cellulose-Based Hybrids Solution. Polymers 2020, 12, 1635.
https://doi.org/10.3390/polym12081635
Regarding safety, the biosafety of nanocellulose was already proved as mentioned in the following
articles.
Mehdi Jorfi, E. Johan Foster. Recent advances in nanocellulose for biomedical applications . J. APPL.
POLYM. SCI. 2015, DOI: 10.1002/APP.41719
Pawan Kumar Mishra, Ondrej Pavelek , Martina Rasticova , Harshita Mishra, Adam Ekielsk
Nanocellulose-Based Biomedical Scaffolds in Future Bioeconomy: A Techno-Legal Assessment of the
State-of-the-Art. Frontier in Bioengineering and Biotechynology. 2022 ,9 , 789603
Also we can conduct economic analysis and biosafety studies in our future studies.
Line 502. Toothpaste contacts with tooth for maximum 4-5 minutes (usually less) and then is removed, and the teeth are thoroughly rinsed with water. Do the authors suppose that for this time Ag/cellulose/F,
an additive, will kill dental bacteria?
Thanks for your valuable comments
The most important property of toothpaste or mouthwash is the rapidity of action, So, regarding almost all
mouth health care products, the extinction time does not exceed minutes to exert the required action.
Furthermore, oral mucosa might act as a temporary reservoir of nanocomposites that contribute to the
sustained release of active ingredients.
https://www.ada.org/en/resources/research/science-and-research-institute/oral-health-topics/toothbrushes

Reviewer 2 Report (New Reviewer)
Comments and Suggestions for Authors
These are my comments and suggestions for improvement of th reviewed paper:
- Can you proof the purity level of the obtained cellulose? How obtained characteristics can be improved through the isolation and purification process? Infromation why did you choose the presented protocol for cellulose obtaining have to be part of the work.
- Better highlights of reasons for using Ulva lactuca have to be added. Also, I have to ask Author to add information about all parameters of drying, crushing and sieving of algae samples in the Materials and methods.
- Add information about using M-S agar, and MRS agar plates and the selected incubation conditions. Maybe it is written in reference 35, but it is necessary to have this information here.
- The VITEK 2 system is not a new system, it has been in the market for more than 2 decades, please rewrite lines 230-233, maybe you can add information about the principle and sample preparations. In the same paragraphs, criteria for colonial selection, incubation conditions for pure cultures as well as long-storage conditions. The whole identification procedure on the Vitek system must be added.
- I am not sure why did Authors used disk-diffusion methods for antimicrobial testing, since the Vitek system can do an antimicrobial susceptibility test in one step using specific cards for that. So, reasons have to be added with all other information about the used methodology.
-line 246: add incubation conditions
line 247: why 0.5 McFarland
- line 250: interpretation of the inhibition zones must be added with the appropriate reference
- In Supplementary materials add all VITEK reports of identification of the presented bacteria.
A better interpretation of Table 5 must be added. Also, instead of 0.0 put nd - not defined; did you do this testing in triplicate? Please add the mean and standard deviation
- Figure 6: did you do this testing in triplicate? Please add pictures of all three repetitions.
Comments on the Quality of English Language
Minor editing of English language required
Author Response
Comments and Suggestions for Authors
These are my comments and suggestions for improvement of the reviewed paper:
- Can you proof the purity level of the obtained cellulose? How obtained characteristics can be improved through the isolation and purification process? Information about why you choose the presented protocol for cellulose obtaining have to be part of the work.
Thanks for reviewer comments
Cellulose was not purified. It was extracted from Ulva lactuca algae, then nanocellulose was prepared. Then by addition of AgNPs, Ag/cellulose nanocomposites was prepared (Figure 1). Af
The chosen protocol was according to (Hamouda et al.,2023).
Hamouda R.A., Fauzia A. K. Qarabai 1 , Fathi S. Shahabuddin 3 , Turki M. Al-Shaikh 1 and Rabab R. Makharita (2023). Antibacterial Activity of Ulva/Nanocellulose and Ulva/Ag/Cellulose Nanocomposites and Both Blended with Fluoride against Bacteria Causing Dental Decay. Polymers 2023, 15, 1047. https://doi.org/10.3390/polym1504104
- Better highlights of reasons for using Ulva lactuca have to be added. Also, I have to ask Author to add information about all parameters of drying, crushing and sieving of algae samples in the Materials and methods.
Thanks for reviewer comments
Ulva is common macro green algae in sea shores of Jedda, Saudia Arabia where study was done. Method of drying was added to the revised manuscript.
- Add information about using M-S agar, and MRS agar plates and the selected incubation conditions. Maybe it is written in reference 35, but it is necessary to have this information here.
Thanks for reviewer comments
Mitis Salivarius Agar is a medium used with supplements for the selective isolation of Streptococcus viridans, such as Streptococcus mitis and Streptococcus salivarius, and Enterococcus faecalis, from specimens containing mixed microbial flora (Takeda et al.,2006).
Takada K, Hayashi K, Sasaki K, Sato T, Hirasawa M. Selectivity of Mitis Salivarius agar and a new selective medium for oral streptococci in dogs. J Microbiol Methods. 2006 Sep;66(3):460-5. doi: 10.1016/j.mimet.2006.01.011
MRS Agar (LMRS) is an enriched selective medium for the isolation and cultivation of Lactobacillus found in clinical, dairy, and food specimens (Sule et al.,2014)..
Süle J, Kõrösi T, Hucker A, Varga L. Evaluation of culture media for selective enumeration of bifidobacteria and lactic acid bacteria. Braz J Microbiol. 2014 Oct 9;45(3):1023-30. doi: 10.1590/s1517-83822014000300035.
It was added to revised manuscript
- The VITEK 2 system is not a new system, it has been in the market for more than 2 decades, please rewrite lines 230-233, maybe you can add information about the principle and sample preparations. In the same paragraphs, criteria for colonial selection, incubation conditions for pure cultures as well as long-storage conditions. The whole identification procedure on the Vitek system must be added.
Thanks for reviewer comments.
the principle, sample preparations criteria for colonial selection, and incubation conditions were added to the revised manuscript.
- I am not sure why did Authors used disk-diffusion methods for antimicrobial testing, since the Vitek system can do an antimicrobial susceptibility test in one step using specific cards for that. So, reasons have to be added with all other information about the used methodology.
The VITEK 2 system can be used in both bacterial identification and antibiotic susceptibility testing.
In the current study we used the The VITEK 2 system in the microbial identification, while we utilized disk diffusion method in antimicrobial susceptibility testing
line 247: why 0.5 McFarland
Thanks for reviewer comments
CLSI recommend to use bacteria with McFarland 0.5 turbidity for antimicrobial testing.
Clinical Laboratory Standards Institute. Available at URL: http://www.clsi.org/
- line 250: interpretation of the inhibition zones must be added with the appropriate reference
Thanks for reviewer comments
Reference was added
- In Supplementary materials add all VITEK reports of identification of the presented bacteria.
Thanks for reviewer comments.
Tables of VITEK results were added to supplementary materials
A better interpretation of Table 5 must be added. Also, instead of 0.0 put nd - not defined; did you do this testing in triplicate? Please add the mean and standard deviation
Thanks for reviewer comments.
Not defined replaced 0, Standard deviation was added
- Figure 6: did you do this testing in triplicate? Please add pictures of all three repetitions.
Thanks for reviewer comments.
Here you are the required figures.

This manuscript is a resubmission of an earlier submission. The following is a list of the peer review reports and author responses from that submission.
Round 1
Reviewer 1 Report
Comments and Suggestions for Authors
Dear Authors of manuscript entitled „Antimicrobial activity of Ag-cellulose nanocomposites against microorganisms isolated from dental caries patients”,
Below please find my comment and suggestions related to your paper:
- Please remove fullstop from the title.
- In abstract Authors says that “This study demonstrated that Ag/cellulose nanocomposites have antibacterial properties against multidrug-resistant bacteria that cause dental caries”. Does this mean that Ag/cellulose has antibacterial properties? What is known about the antibacterial properties of Ag and cellulose separately? It must be clearly stated what is the source of the antimicrobial effect.
- I don't think "VITEK 2" is a good option for a keyword.
- I suggest supplementing the Introduction with indications of caries prevention methods, causes of its formation and treatment methods. Please also add what are the consequences of not treating caries.
- Please refer to the overall oral microflora (not just those that cause dental caries).
Please describe the beneficial effects of native oral microflora and discuss how the test material may influence positive oral microorganisms.
- The Authors write about the antimicrobial effect of nanocellulose, does this material actually have such properties? If so, please describe what they result from? What is the mechanism of action of nanocellulose on gram-positive and gram-negative bacterial cells?
-Why did the authors choose penicillin and augmentin, which is a semi-synthetic drug derived from penicillin, for the study?
-Cotrimoxazole contains two active substances, why did the authors use drugs and not only a single antibiotic? Why have so few antibiotics been tested? Please explain how antibiotics were selected in this study?
- L144 The authors forgot about women.
- Please refer to the standard, publication or relevant isolation protocol to paragraph 2.9.2.
- In Table 4, I suggest omitting the columns containing only "0" and describing this results in the text.
- I suggest that each reference correlated with the manuscript comes from a JCR-listed journal containing an IF.
Unfortunately, without genotypic identification of isolated strains, their taxonomic affiliation cannot be determined. The use of culture media is not a sufficient way to determine species.
Dental caries treatment usually does not involve antibiotic treatment but mechanical removal.
How do the authors want to use the tested material in the treatment of caries?
What is the application correlation of the research undertaken with dentistry?
Comments on the Quality of English LanguageIt is always advisable to have manuscript read by a qualified native speaker before publication.
Author Response
Reviewer 1
Comments and Suggestions for Authors
Dear Authors of manuscript entitled „Antimicrobial activity of Ag-cellulose nanocomposites against microorganisms isolated from dental caries patients”,
Below please find my comment and suggestions related to your paper:
-Please remove full stop from the title.
Thanks for your valuable comments.
The full stop has been removed.
In abstract Authors says that “This study demonstrated that Ag/cellulose nanocomposites have antibacterial properties against multidrug-resistant bacteria that cause dental caries”. Does this mean that Ag/cellulose has antibacterial properties? What is known about the antibacterial properties of Ag and cellulose separately? It must be clearly stated what is the source of the antimicrobial effect.
Thanks for your valuable comments.
The reason of antimicrobial activity of both Ag and cellulose has been added to the revised manuscript.
- I don't think "VITEK 2" is a good option for a keyword.
Thanks for your valuable comments.
VITEK 2 was removed from keywords.
- I suggest supplementing the Introduction with indications of caries prevention methods, causes of its formation and treatment methods. Please also add what are the consequences of not treating caries.
Thanks for your valuable comments.
More details about dental caries were added to revised manuscript.
- Please refer to the overall oral microflora (not just those that cause dental caries).
Please describe the beneficial effects of native oral microflora and discuss how the test material may influence positive oral microorganisms.
Thanks for your valuable comments.
More details about oral microflora were added to revised manuscript.
- The Authors write about the antimicrobial effect of nanocellulose, does this material actually have such properties? If so, please describe what they result from? What is the mechanism of action of nanocellulose on gram-positive and gram-negative bacterial cells?
Thanks for your valuable comments
The antibacterial activity of pure cellulose can only be emphasized in its derivatives, which are created through diverse chemical functionalization with antibacterial groups or by combination with natural or manufactured bioactive components or polymers, as well as metal nanoparticles and metal oxides
-Why did the authors choose penicillin and augmentin, which is a semi-synthetic drug derived from penicillin, for the study?
Thanks for your valuable comments
Even though both penicillin and augmentin belong to beta-lactam antibiotic category, augmentin covers a wide spectrum of gram-positive bacteria and gram-negative coverage in contrast to penicillin. Augmentin is a broad-spectrum antibiotic consisting of a combination of amoxicillin (penicillin-derived) and clavulanic acid (beta-lactamase inhibitor), so it is active against more resistant bacteria (beta-lactamases producer bacteria) and a wide range of infection such as Moraxella catarrhalis, Staphylococcus aureus, lactamase-producing isolates of Haemophilus influenzae etc.
-Cotrimoxazole contains two active substances, why did the authors use drugs and not only a single antibiotic?
Thanks for your valuable comments.
In the Medical field, it is recommended to use sulphonamides combined with trimethoprim to exert their synergistic effect. Cotrimoxazole (TS) is a combination of sulfamethoxazole (SMX)/Trimethoprim (TMP). SMX competitively inhibits dihydrofolate synthase while TMP inhibits dihydrofolate reductase. Inhibition of these two enzymes interferes with two serial steps in folic acid synthesis, thereby inhibiting the synthesis of nucleic acids and proteins in the microorganism.
Synergistic effect of (Trimethoprim and, Sulfamethoxazole) in clotrimazole. Trimethoprim exerts antimicrobial activity by blocking the reduction of dihydrofolate to tetrahydrofolate, Sulfamethoxazole competitively inhibits dihydropteroate synthase, the enzyme responsible for bacterial conversion of PABA to dihydrofolic acid
Why have so few antibiotics been tested? Please explain how antibiotics were selected in this study.
Thanks for your valuable comments.
In this study, we utilized different antibiotics to represent the different classes of antibacterial agents. Furthermore, we focused on the antibiotics that are commonly recommended in the treatment of tooth infections. Penicillin G (PG), ampicillin (AP), and Augmentin (AUG), represent Beta-lactam antibiotic category which are microbial cell wall inhibitors. PG is active with respect to Gram-positive bacteria (staphylococcus, streptococcus, etc. AP displays efficacy against some gram-negative organisms and anaerobic bacteria. AUG is a broad-spectrum antibiotic and a β-lactamase inhibitor. Metronidazole (MZ) belongs to the nitroimidazole antimicrobial class which is a microbial protein synthesis inhibitor and active against most Gram-negative and Gram-positive anaerobic bacteria. Ciprofloxacin (CIP) is a fluoroquinolone that acts by inhibiting nucleic acid synthesis via the blocking of topoisomerase activity. It is active against most Gram-negative bacteria and various Gram-positive bacteria.
In conclusion, we select a presenter member for each antibiotic category which is commonly utilized as dental infection therapy.
- L144 The authors forgot about women.
Thanks for your valuable comments.
Female word was added to the revised manuscript.
- Please refer to the standard, publication, or relevant isolation protocol to paragraph 2.9.2.
Thanks for your valuable comments.
Reference of isolation protocol was added to revised manuscript.
- In Table 4, I suggest omitting the columns containing only "0" and describing this results in the text.
Thanks for your valuable comments.
We have to keep them to demonstrate that fluoride, cellulose, or nanocellulose alone has no antibacterial activity.
- I suggest that each reference correlated with the manuscript comes from a JCR-listed journal containing an IF.
Thanks for your valuable comments.
JCR-listed journal containing IF references were added.
Unfortunately, without genotypic identification of isolated strains, their taxonomic affiliation cannot be determined. The use of culture media is not a sufficient way to determine species.
Thank you for your constructive comment that enhanced our manuscript,
In various studies, the VITEK 2 system is considered a very useful, rapid, and highly automated tool for clinical diagnosis including the identification of gram-negative bacilli from human clinical specimens (1, 2). Chen et al., 2015 reported that the VITEK 2 System is an acceptable microbiological procedure for Kocuria kristinae Identification, which is a gram-positive microbe isolated from pediatric blood specimens. (3) In 2019, the Identification of Candida auris by Use of the VITEK 2 Yeast Identification System in a multi-laboratory evaluation study showed correct identification of Candida auris isolates and that such identification with a high level of confidence, accuracy, and reliability (4)Recently, an accurate positive result was reported in a multicenter study about utilizing the VITEK 2 and VITEK 2 Compact Systems as a new tool to determine MICs in Enterobacterales. (5)
Furthermore, we recommended future work based on a taxonomic affiliation of the isolates depending on your comment.
References
1) Funke G, Monnet D, deBernardis C, von Graevenitz A, Freney J. Evaluation of the VITEK 2 system for rapid identification of medically relevant gram-negative rods. Journal of clinical microbiology. 1998 Jul 1;36(7):1948-52.
2) Gavin PJ, Warren JR, Obias AA, Collins SM, Peterson LR. Evaluation of the Vitek 2 system for rapid identification of clinical isolates of gram-negative bacilli and members of the family Streptococcaceae. European Journal of Clinical Microbiology and Infectious Diseases. 2002 Dec; 21:869-74.
3) Chen HM, Chi H, Chiu NC, Huang FY. Kocuria kristinae: a true pathogen in pediatric patients. Journal of Microbiology, Immunology, and Infection. 2015 Feb 1;48(1):80-4.
4) Ambaraghassi G, Dufresne PJ, Dufresne SF, Vallières É, Muñoz JF, Cuomo CA, Berkow EL, Lockhart SR, Luong ML. Identification of Candida auris by use of the updated Vitek 2 yeast identification system, version 8.01: a multilaboratory evaluation study. Journal of clinical microbiology. 2019 Nov;57(11):10-128.
5) Csiki-Fejer E, Traczewski M, Procop GW, Davis TE, Hackel M, Dwivedi HP, Pincus DH. Multicenter Clinical Performance Evaluation of Omadacycline Susceptibility Testing of Enterobacterales on VITEK 2 Systems. Journal of Clinical Microbiology. 2023 May 10: e00174-23.
Dental caries treatment usually does not involve antibiotic treatment but mechanical removal.
Thanks for your valuable comments.
Systemic antibiotics showed potential efficacy in the prevention or treatment of dental caries according to the following references.
Alaki S. M., Burt B. A., Garetz S. L. The association between antibiotics usage in early childhood and early childhood caries. Pediatric Dentistry. 2009; 31:31–37
Vohra F., Akram Z., Safii S. H., et al. Role of antimicrobial photodynamic therapy in the treatment of aggressive periodontitis: a systematic review. Photodiagnosis and Photodynamic Therapy. 2016; 13:139–147.
How do the authors want to use the tested material in the treatment of caries?
What is the application correlation of the research undertaken with dentistry?
Thanks for your valuable comments.
The application of Ag/cellulose nanocomposites as coating agents for dental implants can be used to significantly improve patients' oral health, leading to their widespread adoption.
Comments on the Quality of English Language
It is always advisable to have manuscript read by a qualified native speaker before publication.
Thanks for reviewer’s valuable comments.
English language was revised using AJE Grammar Check

Reviewer 2 Report
Comments and Suggestions for Authors
The authors carried out a study on Antimicrobial activity of Ag-cellulose nanocomposites against 2 microorganisms isolated from dental caries patients.
The study is very interesting, the methodology is correct but the presentation of the manuscript lacks affinity, it would be interesting if the authors improved the presentation of their scientific data.
Comments:
1- Introduction:
Paragraphs need to be reorganized:
-Present dental caries and their impact on public health,
-Cite the bacteria that cause dental caries and their drug resistance,
- Explain the therapeutic prevention means used at their current level and their limits,
-Present new strategies currently being developed (cite recent publications),
-present your problem and argue your choice of model used.
Materials and methods:
Chemicals:
-List all chemicals used in the study including culture media.
*Note : de Man.
-Were the algae samples preserved before use? if yes, how and at what temperature?
Synthesis of nanocellulose:
-how the mixture was filtered and using what type of materials (size of the filtration ports).
*It is necessary to better explain any syntheses carried out.
Collection and isolation of bacterial strains:
-Add female to 32%. What is the point of specifying the percentage relating to gender? have you carried out statistical tests?
*de man => de Man
Identification of bacterial isolates:
-Explain all abbreviations used.
-Explain the principle of VITEK, bacterial identification via biochemical characteristics or antibiotic resistance?
Antimicrobial susceptibility tests:
-Explain the abbreviation used.
Antibacterial Activities of cellulose, AgNPs, nanocellulose, and Ag/cellulose nano-167 composites with and without fluoride against isolated bacteria:
-Detail how the technique was carried out.
Results and discussion:
-The same abbreviations for bacteria must be used throughout the manuscript.
-Discuss the results from table 3 relating to part 3.3.
-In table 4, add the concentrations of the products used for anti-bacterial treatment.
-In the text, add the concentration of your synthesis which has anti-bacterial activity.
- P. aeruginosa in italics.
-"Liao et al.,[58] showed that AgNPs have antibacterial properties against multidrug-resistant P. aeruginosa in a concentration and time-dependent manner". correct this sentence and specify at what concentration it had an antibacterial effect, it is better to compare with your results.
Comments on the Quality of English LanguageThe text presentation format must be corrected:
-The punctuation,
-The spaces between words and especially between numbers in the results section.
-correct English mistakes.
Author Response
Reviewer 2
Comments and Suggestions for Authors
The authors carried out a study on Antimicrobial activity of Ag-cellulose nanocomposites against 2 microorganisms isolated from dental caries patients.
The study is very interesting, the methodology is correct but the presentation of the manuscript lacks affinity, it would be interesting if the authors improved the presentation of their scientific data.
Comments:
1- Introduction:
Paragraphs need to be reorganized:
-Present dental caries and their impact on public health,
-Cite the bacteria that cause dental caries and their drug resistance,
- Explain the therapeutic prevention means used at their current level and their limits,
-Present new strategies currently being developed (cite recent publications),
-present your problem and argue your choice of model used.
Thanks for your valuable comments.
More organized and new information about dental carries and its treatment was added to the revised manuscript.
Materials and methods:
Chemicals:
-List all chemicals used in the study including culture media.
Thanks for your valuable comments.
Chemicals used were added to the revised manuscript.
*Note: de Man.
Thanks for your valuable comments
Corrected
-Were the algae samples preserved before use? If yes, how and at what temperature?
Alga was used directly after drying, and the remaining algae was preserved at ambient temperature until use.
Synthesis of nanocellulose:
-how the mixture was filtered and using what type of materials (size of the filtration ports).
The mixture was filtered by Whatman® qualitative filter paper, Grade 1 (Merck).
*It is necessary to better explain any syntheses carried out.
Thanks for your valuable comments.
All synthesis details were added.
Collection and isolation of bacterial strains:
-Add female to 32%. What is the point of specifying the percentage relating to gender? have you carried out statistical tests?
Thanks for your valuable comments.
Gender was removed from the revised manuscript as I found it not necessary, so no need to statistical analysis.
*de man => de Man
Thanks for your valuable comments
Corrected
Identification of bacterial isolates:
-Explain all abbreviations used.
Thanks for your valuable comments
Abbreviations were added
-Explain the principle of VITEK, bacterial identification via biochemical characteristics or antibiotic resistance?
Thanks for your valuable comments
The principle of VITEK, bacterial identification via biochemical characteristics was added to the revised manuscript.
Antimicrobial susceptibility tests:
-Explain the abbreviation used.
Thanks for your valuable comments
Abbreviations were added
Antibacterial Activities of cellulose, AgNPs, nanocellulose, and Ag/cellulose nanocomposites with and without fluoride against isolated bacteria:
-Detail how the technique was carried out.
Thanks for your valuable comments
Technique was added
Results and discussion:
-The same abbreviations for bacteria must be used throughout the manuscript.
Thanks for your valuable comments.
Abbreviations were unified.
-Discuss the results from table 3 relating to part 3.3.
Thanks for your valuable comments
More discussion was added
-In table 4, add the concentrations of the products used for anti-bacterial treatment.
Thanks for your valuable comments
Done
-In the text, add the concentration of your synthesis which has anti-bacterial activity.
Thanks for your valuable comments
Done
- P. aeruginosa in italics.
Thanks for your valuable comments
Corrected.
-"Liao et al.,[58] showed that AgNPs have antibacterial properties against multidrug-resistant P. aeruginosa in a concentration and time-dependent manner". correct this sentence and specify at what concentration it had an antibacterial effect, it is better to compare with your results.
Thanks for your valuable comments
Done.
Comments on the Quality of English Language
The text presentation format must be corrected:
-The punctuation,
-The spaces between words and especially between numbers in the results section.
-correct English mistakes.
Thanks for reviewer’s valuable comments.
English language was revised using AJE Grammar Check

Reviewer 3 Report
Comments and Suggestions for Authors
The article "Antimicrobial Activity of Ag-Cellulose Nanocomposites Against Microorganisms Isolated from Dental Caries Patients" explores a subject that has garnered considerable attention in recent decades: nanocomposites of Ag and (nano)cellulose. Silver-based polymer composites have consistently demonstrated excellent antibacterial and antifungal properties.
The manuscript, while containing a few typographical errors, is written in a comprehensive and reader-friendly language, making it relatively easy to understand. The concept of applying Ag/cellulose in the field of dentistry is intriguing, and the proposed method of synthesizing the composites through the reduction of silver using algae extracts is both timely and innovative.
However, the article requires substantial improvements across all its sections. Despite the potential, I believe it should be published only after a total revision. The part related to chemistry is unfortunately weak. The authors didn’t prove that they really obtained nanocellulose. 10 minutes in hot HCl solution does not necessarily result in NC. It is powder cellulose, not nanocellulose will be obtained. Physico-chemical methods must be discussed properly.
I offer the following notes in the hope that they will assist the authors in enhancing the quality of the manuscript.
Line 49. Chemical inertness of nanocellulose (which is just another scale option of cellulose) is doubtable. This statement of [3] contradicts the following paragraph (and existing data) about formation of nanocomposites, which are not just mixtures of NC and metals: "polymer acts as a surface capping, reducing, and/or stabilizing agent". Due to what? Due to the OH-groups and the 3D network of cellulose, saying in simple words. For instance, see the mechanism proposed in https://doi.org/10.3390/gels9050390, when the coordination of metal is shown due to these functional groups.
Line 66-67. This sentence is not informative. What NP? What many cases? Please, clarify the point.
Line 67. NC does not perform antibacterial properties itself; it can only be a matrix for antibacterial agents. In [12] such agent was chitosan. This sentence and the next one should be paraphrased to not confuse the reader.
Line 70. Hydroxyl groups input into the reactivity of cellulose, as well as in the formation of hydrogen bonds, but not in a formation of any new structures. In water, NC form a suspension, not a structure.
Line 70-71. Nanocomposites is a very broad term, not all nanocomposites are antibacterial. Please, specify.
Line 70. What did the authors mean on stable structure of cellulose in water?
Line 71. Add “For example,” when...
Interestingly why the authors give the example about Zn and ZnO NP in the article related to Ag NP.
Line 73. AgCl-reinforced? Check the reference [16].
Lines 63-75. I recommend revising this paragraph totally. It is illogical now. The last sentence is not "also", it is about the source, while in the paragraph the properties are characterized. Since algae is not so common but fascinating source, it definitely deserves one more paragraph about state-of-the-art in the cellulose production from algae. Phyto-chemical reduction needs a mention as well.
Line 78. Antibacterial effect of AgNO3?
Line 78. There was not a word about fluoride above. I guess why it is in agenda of the authors who write about dental care, but, please, explain it in the introduction.
Line 82. There were other chemicals, e.g., AgNO3, HCl, etc. I suppose, that antibiotic susceptibility discs and cards are not really chemicals. Probably, they may be called antimicrobial materials or there is another classification for them? I think the heading of 2.1 should be corrected accordingly.
Line 84. The last sentence in 2.1 is incomplete (there is no verb).
Line 86. Was there an assessment of the species, once the algae were not purchased? Usually, a professional botanist should confirm the specie.
Line 91. Please, describe the procedure here in brief.
Line 93. what was concentration of HCl? What was the ratio cellulose to the acid solution?
Line 94. How did you provide filtration? What type of filter was used? Wasn't nano-fraction of cellulose lost with the filtrate, passing through the filter?
Line 96. What was the purpose of drying? Cellulose does not "like" heating, it leads to the structural changes. Not that big due to 60 C, but it happens.
Line 97. Please provide the experimental flowchart. It is unclear now why you produced NC (as you suppose), then you prepare Ag NP with algae (not cellulose extracted from algae, not NC produced from those cellulose). What happened to the Ag NP from sub-chapter 2.5?
Line 99. Please, provide the concentration of the solution rather than the weigh.
Line 101. Did you remove algae from the suspension before the following application?
Line 104. What happened next? Was it washed? dried? How? or used as suspension?
Line 107. How was counted 1.7 mg/ml? You added 0.17 g of silver nitrate (which is not the same as silver) to 90 ml of water. It is not 1.7 mg/ml
Line 107. Please name all the materials consistently throughout the text. Is ulva/nanocellulose the same as nanocellulose from 2.1? And Ulva/Ag/cellulose nanocomposites is the same as Ag/cellulose nanocomposites from 2.6? It is not very clear now.
Line 108. What is fluoride? Do you mean an ion? It must be described in 2.1. What means 1.23% of fluoride? Is it the concentration in the resulting suspension of one ml of 1.7 mg/mL silver nanoparticles, 4 mg/mL Ulva/ nanocellulose, and 2 mg/mL Ulva/Ag/cellulose nanocomposites?
Line 112. Not chemical functional groups, just functional groups.
Line 144. 32% females?
Lines 148-149, check the sentence.
Line 160. Decrypt abbreviations when use them for the first time.
Line 169. In the aim of the study the antibacterial activity of AgNO3 was also claimed.
Lines 169-171. The verb is missed in this sentence.
Line 174. The FT-IR spectroscopy analysis (3.1) in the article requires revision. Section 2.6 was unclear, leaving me with questions regarding the origin of NH2 groups, amide I, amide II, and PO2 groups in the Ag/cellulose nanocomposite. It is essential to clarify whether these components originate from the algae extract, since the extract was mentioned to be used as a reducer. However, the authors have not provided information about the extract's composition, making it challenging for readers to grasp how NH2 and PO2 groups are present in cellulose. Clarification on this matter as well as the relevant examples from the literature is necessary.
Line 189. Now it does not look very "common" for cellulose-Ag composite. Check the lattice planes of the peaks and provide the information which peaks you addressed to cellulose and which to Ag.
Additionally, the crystallinity and sizes of the crystals were claimed in the experimental part. However, I do not see the data in the results. It's a pity because how else you can confirm the presence of nanocellulose in the composites?
Line 196. The origin of Na, Mg, Si, Cl, Ca, Fe, Cu, and Zn should be discussed. How did they appear in the Ag-cellulose composites?
Line 209. Unfortunately, I do not see any spherical silver nanoparticles (probably, because of the scale) in Fig. 2A.
Line 220. I suppose that the catalytic activity is not relevant to the study on antimicrobial properties.
Lines 221-222. How can you prove that those are Ag NP, having numerous other elements, such as Zn, Ca, Fe, Na, etc, in the composite (section 3.3).
Line 225. This is what i was saying above. How is possible to find 10-60 nm particles in the Fig 1, if 200 microns is a third part of the image?
Lines 227-230. Avoid discussion, e.g., "shows well-organized by-layer porosity architecture and substantial surface area ", in the name of the figures, just point what is depicted. In the case of microscopy, it is better to note the magnification.
Line 233. Typo: Table 1. Eight
Tables 1 and 2. Probably, it is better to provide tables 1 and 2 as supplementary information, since they present not properties, but routine identification. Also, it seems, that a right part of table 1 is missed ecause of the formatting.
Lines 266-301. Is it necessary to know from which patients the bacteria were isolated? I'm asking because I am a chemist, not a microbiologist, and I'm unsure of the relevance of this information. If, for instance, the authors had drawn conclusions such as colonies isolated from, let's say, patients 12, 14, and 30 showed resistance to all types of antibiotics while others did not (or something like this), it would provide a clearer understanding. However, for now, there is no conclusion after this part.
Additionally, if the authors believe that the description provided in lines 266-301 is crucial for the article, I would suggest presenting this information in a tabular format. In its current textual form, it can be quite challenging to follow all the numbers and species, and a table would greatly improve readability and comprehension.
Line 268. Typo: 10 10,62,39
Lines 317-318. Cellulose/fluoride, nanocellulose/fluoride, AgNPs, AgNPs /fluoride, and Ag/cellulose nanocomposites and Ag/cellulose nanocomposites /fluoride is not what I can see in Table 4.
But the only question is why did you conclude that Cellulose/fluoride, nanocellulose/fluoride are antibacterial. I see zeros in the inhibition zones diameters for them.
Table 4. Turn the headings in the Table to read the words from the bottom to up (as it is usually written). I see that several words are missed in the line with the headings of the columns.
Figure 3. What is MZ?
Line 330. A paragraph above it was written: "Cellulose/fluoride, nanocellulose/fluoride [...] possessed antibacterial activity against isolated bacteria", in the table there are 0.00 cm for cellulose, fluoride and nanocellulose, which mean that they do not have antibacterial properties, and in lines 331 and 332 it is written that they are antibacterial due to fluoride. These three statements are controversial.
Line 338. Exactly! This is what I am writing from the very beginning of this manuscript.
Line 377. “Ag/cellulose nanocomposites outperformed nanocellulose” Sure, because nanocellulose is not antibacterial...
Reference list. DOI is needed to all the references.
I advise re-submission of the manuscript after revision.
Author Response
Comments and Suggestions for Authors
The article "Antimicrobial Activity of Ag-Cellulose Nanocomposites Against Microorganisms Isolated from Dental Caries Patients" explores a subject that has garnered considerable attention in recent decades: nanocomposites of Ag and (nano)cellulose. Silver-based polymer composites have consistently demonstrated excellent antibacterial and antifungal properties.
The manuscript, while containing a few typographical errors, is written in a comprehensive and reader-friendly language, making it relatively easy to understand. The concept of applying Ag/cellulose in the field of dentistry is intriguing, and the proposed method of synthesizing the composites through the reduction of silver using algae extracts is both timely and innovative.
However, the article requires substantial improvements across all its sections. Despite the potential, I believe it should be published only after a total revision. The part related to chemistry is unfortunately weak. The authors didn’t prove that they really obtained nanocellulose. 10 minutes in hot HCl solution does not necessarily result in NC. It is powder cellulose, not nanocellulose will be obtained. Physico-chemical methods must be discussed properly.
I offer the following notes in the hope that they will assist the authors in enhancing the quality of the manuscript.
Line 49. Chemical inertness of nanocellulose (which is just another scale option of cellulose) is doubtable. This statement of [3] contradicts the following paragraph (and existing data) about formation of nanocomposites, which are not just mixtures of NC and metals: "polymer acts as a surface capping, reducing, and/or stabilizing agent". Due to what? Due to the OH-groups and the 3D network of cellulose, saying in simple words. For instance, see the mechanism proposed in https://doi.org/10.3390/gels9050390, when the coordination of metal is shown due to these functional groups.
Thanks for your valuable comments!
The sentence of chemical inertness was removed.
Line 66-67. This sentence is not informative. What NP? What many cases? Please, clarify the point.
Thanks for your valuable comments!
The sentence has been modified in the revised manuscript.
Line 67. NC does not perform antibacterial properties itself; it can only be a matrix for antibacterial agents. In [12] such agent was chitosan. This sentence and the next one should be paraphrased to not confuse the reader.
Thanks for your valuable comments!
The sentence has been rephrased in the revised manuscript.
Line 70. Hydroxyl groups input into the reactivity of cellulose, as well as in the formation of hydrogen bonds, but not in a formation of any new structures. In water, NC form a suspension, not a structure.
Thanks for your valuable comments!
The sentence has been rephrased in the revised manuscript.
Line 70-71. Nanocomposites is a very broad term, not all nanocomposites are antibacterial. Please, specify.
Thanks for your valuable comments!
The sentence has been modified to be more clear..
Line 70. What did the authors mean on stable structure of cellulose in water?
Thanks for your valuable comments!
The sentence has been rephrased in the revised manuscript.
Line 71. Add “For example,” when...
Interestingly why the authors give the example about Zn and ZnO NP in the article related to Ag NP.
Thanks for your valuable comments!
I give it as another example of nanocomposite containing metal.
Line 73. AgCl-reinforced? Check the reference [16].
Thanks for your valuable comments!
Reference was checked.
Lines 63-75. I recommend revising this paragraph totally. It is illogical now. The last sentence is not "also", it is about the source, while in the paragraph the properties are characterized. Since algae is not so common but fascinating source, it definitely deserves one more paragraph about state-of-the-art in the cellulose production from algae. Phyto-chemical reduction needs a mention as well.
Thanks for your valuable comments!
More information was added to revised manuscript.
Line 78. Antibacterial effect of AgNO3?
Thanks for your valuable comments!
AgNPs replaced AgNO3
Line 78. There was not a word about fluoride above. I guess why it is in agenda of the authors who write about dental care, but, please, explain it in the introduction.
Thanks for your valuable comments!
Fluoride importance was added to revised manuscript.
Line 82. There were other chemicals, e.g., AgNO3, HCl, etc. I suppose, that antibiotic susceptibility discs and cards are not really chemicals. Probably, they may be called antimicrobial materials or there is another classification for them? I think the heading of 2.1 should be corrected accordingly.
Thanks for your valuable comments!
More chemicals were added. Materials was used instead of chemicals.
Line 84. The last sentence in 2.1 is incomplete (there is no verb).
Thanks for your valuable comments!
Modified.
Line 86. Was there an assessment of the species, once the algae were not purchased? Usually, a professional botanist should confirm the specie.
The algae was identified by Prof: Ragaa Hamouda (Professor of Microbiology, University of Sadat City, Egypt according to .
Taylor, W.R. Marine algae of the eastern tropical and subtropical coasts of the Americas, University of Michigan Scientific series volume XXI, 1985, United states of Amerrica,
Line 91. Please, describe the procedure here in brief.
Thanks for your valuable comments!
The procedure was described in details.
Line 93. what was concentration of HCl? What was the ratio cellulose to the acid solution?
HCl Concentrations was 38%, about 40 gm of cellulose was added to 200 ml acid solution (2W:5V)
Line 94. How did you provide filtration? What type of filter was used? Wasn't nano-fraction of cellulose lost with the filtrate, passing through the filter?
Filtration was done using filter paper Whatman® qualitative filter paper, Grade 1 (Merck). It will not be lost as it is cellulose not nanocellulose..
Line 97. Please provide the experimental flowchart. It is unclear now why you produced NC (as you suppose), then you prepare Ag NP with algae (not cellulose extracted from algae, not NC produced from those cellulose). What happened to the Ag NP from sub-chapter 2.5?
Flowchart was added to the revised manuscript.
Line 101. Did you remove algae from the suspension before the following application?
The silver nanoparticle pellets were washed 3 times to remove algal residue and were oven-dried at 50 °C for 3 h.
Line 104. What happened next? Was it washed? dried? How? or used as suspension?
The mixture was centrifuged at 8000 rpm at 30 °C for 30 min. The supernatant was discarded. The Ag/cellulose nanocomposites were washed 3 times after dried to remove algal residue and was oven-dried at 50 °C for 3 h
Line 107. How was counted 1.7 mg/ml? You added 0.17 g of silver nitrate (which is not the same as silver) to 90 ml of water. It is not 1.7 mg/ml
After preparation of AgNPs, Dried AgNPs was added to distilled water to form suspension at concentration of 1.7 mg/ml to be used in nanocomposite preparation.
Line 107. Please name all the materials consistently throughout the text. Is ulva/nanocellulose the same as nanocellulose from 2.1? And Ulva/Ag/cellulose nanocomposites is the same as Ag/cellulose nanocomposites from 2.6? It is not very clear now.
Thanks for your valuable comments!
Yes, they are the same. The names were unified.
Line 108. What is fluoride? Do you mean an ion? It must be described in 2.1. What means 1.23% of fluoride? Is it the concentration in the resulting suspension of one ml of 1.7 mg/mL silver nanoparticles, 4 mg/mL Ulva/ nanocellulose, and 2 mg/mL Ulva/Ag/cellulose nanocomposites?
Thanks for your valuable comments.
Fluoride was added as Hama fluoride topical gel (1.23% fluoride ion) Kal-AlHamaya , Saudia Arabia , Antibiotic. I t was added to section 2.1 and section 2.7
Line 112. Not chemical functional groups, just functional groups.
Thanks for your valuable comments!
corrected.
Line 144. 32% females?
Thanks for your valuable comments!
Male and female gender was removed from revised manuscript.
Lines 148-149, check the sentence.
Thanks for your valuable comments!
Checked
Line 160. Decrypt abbreviations when use them for the first time.
Thanks for your valuable comments!
Checked
Lines 169-171. The verb is missed in this sentence.
Thanks for your valuable comments!
Checked
Line 174. The FT-IR spectroscopy analysis (3.1) in the article requires revision. Section 2.6 was unclear, leaving me with questions regarding the origin of NH2 groups, amide I, amide II, and PO2 groups in the Ag/cellulose nanocomposite. It is essential to clarify whether these components originate from the algae extract, since the extract was mentioned to be used as a reducer. However, the authors have not provided information about the extract's composition, making it challenging for readers to grasp how NH2 and PO2 groups are present in cellulose. Clarification on this matter as well as the relevant examples from the literature is necessary.
Thanks for your valuable comments!
All peaks and discussion were checked in the revised manuscript.
Line 189. Now it does not look very "common" for cellulose-Ag composite. Check the lattice planes of the peaks and provide the information which peaks you addressed to cellulose and which to Ag.
Additionally, the crystallinity and sizes of the crystals were claimed in the experimental part. However, I do not see the data in the results. It's a pity because how else you can confirm the presence of nanocellulose in the composites?
Thanks for your valuable comments
The peaks of Ag was differentiated from cellulose. The sharpness of the diffraction peaks indicates the high crystallization performance of silver
Line 196. The origin of Na, Mg, Si, Cl, Ca, Fe, Cu, and Zn should be discussed. How did they appear in the Ag-cellulose composites?
The most predominant elements are C, O, and Ag, that had percentage weight 33, 32, and 8 %
So the cellulose appear as present C, and O, and Ag to form nanocomposites, the trace elements appear may be absorbed by algae and not dissolved during the nanocellulose preparations
Line 209. Unfortunately, I do not see any spherical silver nanoparticles (probably, because of the scale) in Fig. 2A.
Thanks for your valuable comments!
polydispersed hexagonal-shaped nanoparticles
Line 220. I suppose that the catalytic activity is not relevant to the study on antimicrobial properties.
Thanks for your valuable comments!
Changed to antibacterial activity
Line 225. This is what i was saying above. How is possible to find 10-60 nm particles in the Fig 1, if 200 microns is a third part of the image?
Thanks for your valuable comments!
100 nm is scale of TEM, while 200 µm is scale of SEM. Our particle size is from 12.37 to 19.12 nm
Lines 227-230. Avoid discussion, e.g., "shows well-organized by-layer porosity architecture and substantial surface area ", in the name of the figures, just point what is depicted. In the case of microscopy, it is better to note the magnification.
Thanks for your valuable comments!
Modified
Line 233. Typo: Table 1. Eight
Thanks for your valuable comments!
Corrected
Tables 1 and 2. Probably, it is better to provide tables 1 and 2 as supplementary information, since they present not properties, but routine identification. Also, it seems, that a right part of table 1 is missed ecause of the formatting.
Thanks for your valuable comments!
I preferred to put them in the main text to discuss the isolated strains which we will try antibacterial activities of antibiotics and nanocomposites on it. I modified Table 1
Lines 266-301. Is it necessary to know from which patients the bacteria were isolated? I'm asking because I am a chemist, not a microbiologist, and I'm unsure of the relevance of this information. If, for instance, the authors had drawn conclusions such as colonies isolated from, let's say, patients 12, 14, and 30 showed resistance to all types of antibiotics while others did not (or something like this), it would provide a clearer understanding. However, for now, there is no conclusion after this part
Thanks for your valuable comments!
The number of patient isolate is necessary to be mentioned as all the following results are related to patient number.
Additionally, if the authors believe that the description provided in lines 266-301 is crucial for the article, I suggest presenting this information in a tabular format. In its current textual form, it can be quite challenging to follow all the numbers and species, and a table would greatly improve readability and comprehension.
Thanks for your valuable comments!
I tried to put it on tabular format but found it will be more complex.
Line 268. Typo: 10 10,62,39
Thanks for your valuable comments!
Corrected
Lines 317-318. Cellulose/fluoride, nanocellulose/fluoride, AgNPs, AgNPs /fluoride, and Ag/cellulose nanocomposites and Ag/cellulose nanocomposites /fluoride is not what I can see in Table 4.But the only question is why did you conclude that Cellulose/fluoride, nanocellulose/fluoride are antibacterial. I see zeros in the inhibition zones diameters for them.
Thanks for your valuable comments.
I mean that although they have some zero with some isolates, they still have inhibition zones with other isolates.
Table 4. Turn the headings in the Table to read the words from the bottom to up (as it is usually written). I see that several words are missing in the line with the headings of the columns.
Thanks for your valuable comments.
Done
Figure 3. What is MZ?
Thanks for your valuable comments.
MZ is metronidazole
Line 330. A paragraph above it was written: "Cellulose/fluoride, nanocellulose/fluoride [...] possessed antibacterial activity against isolated bacteria", in the table there are 0.00 cm for cellulose, fluoride and nanocellulose, which mean that they do not have antibacterial properties, and in lines 331 and 332 it is written that they are antibacterial due to fluoride. These three statements are controversial.
Thanks for your valuable comments.
None of the isolated bacteria displayed an inhibitory zone of fluoride, cellulose, or nanocellulose alone but addition of fluoride to both cellulose and nanocellulose will have antibacterial activity
Reference list. DOI is needed for all the references.
Thanks for your valuable comments.
DOI was added.

Round 2
Reviewer 1 Report
Comments and Suggestions for Authors
Dear Authors, after reading your proofed version of manuscript, I still have some doubts. Authors wrote: "The antibacterial activity of pure cellulose can only be emphasized in its derivatives, which are created through diverse chemical functionalization with antibacterial groups or by combination with natural or manufactured bioactive components or polymers, as well as metal nanoparticles and metal oxides".
It is still not crear pointed IF PURE NANOCELLULOSE SHOWS ANTIBACTERIAL ACTIVITY OR NOT.
In my opinion this material could be a carrier for some antibacterial agents but authors wrote that pure cellulose (without any additives) can show antimicrobial effect which is not true I think. It must be clearly pointed, please rewrite this information.
I maintain my comment that strains should be characterized at least for MLST testing.
Authors wrote: "Systemic antibiotics showed potential efficacy in the prevention or treatment of dental caries according to the following references." Are the authors sure that antibiotics can be PREVENTIVELY used in the treatment of caries? What would that look like? I insist that caries is not treated with antibiotics but with other, especially mechanical methods. The antibiotics can be administered, but not into the body (not swallow, not intravenously), only during mechanical removal, the medicine can be placed in the treated tooth, but the authors did not write about this.
"The application of Ag/cellulose nanocomposites as coating agents for dental implants can be used to significantly improve patients' oral health, leading to their widespread adoption" - Please provide a reference to this conclusion. I have the impression that it could have been a different cellulose than that described in the study.
Comments on the Quality of English LanguageIt is always good practise to check the English by professional editor.
Author Response
Comments and Suggestions for Authors
Dear Authors, after reading your proofed version of manuscript, I still have some doubts.
1)Authors wrote: "The antibacterial activity of pure cellulose can only be emphasized in its derivatives, which are created through diverse chemical functionalization with antibacterial groups or by combination with natural or manufactured bioactive components or polymers, as well as metal nanoparticles and metal oxides".
It is still not clear pointed IF PURE NANOCELLULOSE SHOWS ANTIBACTERIAL ACTIVITY OR NOT.
In my opinion this material could be a carrier for some antibacterial agents but authors wrote that pure cellulose (without any additives) can show antimicrobial effect which is not true I think. It must be clearly pointed, please rewrite this information.
Thanks for your valuable comment.
Pure cellulose has no antibacterial activities, however addition of fluoride or Ag + is necessary for antibacterial activities while nanocellulose carriers improve the efficacy of fluoride or Ag+
In Line 54-55, we mentioned that a nanocomposite is formed when nanocellulose and metal nanoparticles are joined by using cellulose as a soft matrix to hold inorganic fillers such as metal nanoparticles.
Also it is shown in table 5 that cellulose and nanocellulose has no antibacterial activities. Only nanocellulose combined with fluoride or AgNO3 has antibacterial activities.
Furthermore, depending on your request we deleted line 88: Much emphasis has been given to the use of nanocellulose as an antibacterial substance.
2)I maintain my comment that strains should be characterized at least for MLST testing.
Thanks for your valuable comment,
Thank you for your constructive comment that enhanced our research, for sure we will take your comment in consideration in any further work as we recommend microbial identification based on a taxonomic affiliation of the isolates.
3)Authors wrote: "Systemic antibiotics showed potential efficacy in the prevention or treatment of dental caries according to the following references." Are the authors sure that antibiotics can be PREVENTIVELY used in the treatment of caries? What would that look like? I insist that caries is not treated with antibiotics but with other, especially mechanical methods. The antibiotics can be administered, but not into the body (not swallow, not intravenously), only during mechanical removal, the medicine can be placed in the treated tooth, but the authors did not write about this.
Thanks for your valuable comments
Antibiotics are used in the treatment of various dental conditions including dental abscesses periodontal infections, and acute necrotizing ulcerative gingivitis (ANUG) which may arise due to an overgrowth of oral normal bacteria. So the treatment of such infections might inhibit the development of tooth decay that is caused as a result of bacterial acid production.
For more clarification, depending on your request in line no 84, we replaced (dental caries by periodontal infection)
4)"The application of Ag/cellulose nanocomposites as coating agents for dental implants can be used to significantly improve patients' oral health, leading to their widespread adoption" - Please provide a reference to this conclusion. I have the impression that it could have been a different cellulose than that described in the study
Thanks for your valuable comments
The application of Ag/cellulose nanocomposites as coating agents for dental implants according to the following reference.
Zafar, M.S.; Fareed, M.A.; Riaz, S.; Latif, M.; Habib, S.R.; Khurshid, Z. Customized Therapeutic Surface Coatings for Dental Implants. Coatings 2020, 10, 568. https://doi.org/10.3390/coatings10060568
Bolenwar A, Reche A, Dhamdhere N, Rathi S. Applications of Silver Nanoparticles in Dentistry. Cureus. 2023 Aug 25;15(8):e44090. doi: 10.7759/cureus.44090. PMID: 37750112; PMCID: PMC10518072.

Reviewer 2 Report
Comments and Suggestions for Authors
The requested modifications are made, the introduction could be improved. The format, puntuation and English to be reviewed before publication.
Comments on the Quality of English Languageto review before publication.
Author Response
Comments and Suggestions for Authors
The requested modifications are made, the introduction could be improved. The format, puntuation and English to be reviewed before publication.
Thanks for your valuable comments
The introduction has been improved.
English was revised.
Reviewer 3 Report
Comments and Suggestions for Authors
The manuscript was improved by the authors, however, the major questions stayed the same.
The biggest one is how did the authors prove that they obtained nanocellulose, not a powder or microcrystalline cellulose? Which method they applied for that?
The second one. I still do not see the answer how did the authors explain the presence of amides in cellulose. Before the explanation about how they interact with Ag+, it would be good to explain what is their origin. There is no N in cellulose macromolecule. The same in line 259.
Line 261-262. What is the origin of C≡C-H and C-S groups in cellulose?
Line 180 : "Scherrer's equation was used to calculate the cellulose size". Lines 263-272 : There are no results of the "cellulose size" calculation. Yes, indeed, it has an impact on the patterns. But if you claim that you calculated it, provide the results.
The planes for cellulose in Fig. 2b are unusual, as i wrote before. Check them.
I do not see any nanoparticles in Fig 3a, to which you refer in line 289. There are neither spherical nor hexagonal NP in it. The scale is too big to see any NP. You were discussing Fig. 3a there, you wrote: ".... face area (Figure 3 A). Additionally, it showed that polydispersed hexagonal-shaped nanoparticles were present and distributed throughout the composite film". No, figure 3a does not show that.
Lines 426-430. Cellulose+fluoride and NC+fluoride showed antibacterial activity? It is very strange, to the best of my knowledge. I suppose that you should support your position. Probably, you can provide examples from literature other than your own article in Polymers in line 436? Those examples which are given in [32] are not related to the issue. Graphene oxide and CaO have completely different chemical nature than fluoride.
Minor comments:
Line 100. The symbols @ should be replaced with / or +.
Line 121. NaOH, ethanol and HCl are not materials, they are chemicals. I was saying about discs and cards that they are materials, not chemicals.
Line 128. Probably it should be a comma before ID-Gram-Positive...?
Line 146. Is there a space between Na and OH?
Line 250. It is better to replace the word peaks with absorption bands (or just bands) for FTIR in the text.
Line 278. The answer for my previous question about EDS is better to insert into the manuscript: "So the cellulose appear as present C, and O, and Ag to form nanocomposites, the trace elements appear may be absorbed by algae and not dissolved during the nanocellulose preparations".
Table 4, instead of writing 0.4 mg/mL in every cell, you would better write it in the name of the Table for not repeating it many times.
Line 429. Correct the commas/fullstops/spaces: No 48,8,40, and 25. 10.
Author Response
Comments and Suggestions for Authors
The manuscript was improved by the authors, however, the major questions stayed the same.
- The biggest one is how did the authors prove that they obtained nanocellulose, not a powder or microcrystalline cellulose? Which method did they apply for that?
Thanks for valuable comments.
From TEM analysis, it is shown that nanoparticles are with diameters ranging from 12.37 to 19.12 nm. also by XRD, we added the size of nanoparticles obtained by XRD in the manuscript
2)The second one. I still do not see the answer to how the authors explained the presence of amides in cellulose. Before the explanation about how they interact with Ag+, it would be good to explain what their origin is. There is no N in cellulose macromolecule. The same in line 259.Line 261-262. What is the origin of C≡C-H and C-S groups in cellulose?
Thanks for valuable comments.
Extract of green alga is rich in amide carboxylic, and nitro compounds which were used for the synthesis of spherical AgNPs by catalyzing the reduction of silver ions (Ag+) to Ag0 as supported by the following references, and the same peak are present in the following references
[43] Hamouda, R. A.; Abd El Maksoud, A. I.; Wageed, M.; Alotaibi, A. S.; Elebeedy, D.; Khalil, H.; Abdella, A. Characterization and anticancer activity of biosynthesized Au/cellulose nanocomposite from Chlorella vulgaris. Polymers, 2021, 13(19), 3340
[44] Madhiyazhagan P., Murugan K., Kumar A.N., Nataraj T., Subramaniam J., Chandramohan B., Panneerselvam C., Dinesh D., Suresh U., Nicoletti M. One pot synthesis of silver nanocrystals using the seaweed Gracilaria edulis: Biophysical characterization and potential against the filariasis vector Culex quinquefasciatus and the midge Chironomus circumdatus. J. Appl. Phycol. 2017;29:649–659.
3)Line 180 : "Scherrer's equation was used to calculate the cellulose size". Lines 263-272 : There are no results of the "cellulose size" calculation. Yes, indeed, it has an impact on the patterns. But if you claim that you calculated it, provide the results.
The planes for cellulose in Fig. 2b are unusual, as i wrote before. Check them.
Thanks for valuable comments
Checked, and corrected
4)I do not see any nanoparticles in Fig 3a, to which you refer in line 289. There are neither spherical nor hexagonal NP in it. The scale is too big to see any NP. You were discussing Fig. 3a there, you wrote: ".... face area (Figure 3 A). Additionally, it showed that polydispersed hexagonal-shaped nanoparticles were present and distributed throughout the composite film". No, figure 3a does not show that.
Thanks for valuable comments
Sorry it was written by mistake as it belongs to Figure 3B (TEM). I removed it from SEM.
5)Lines 426-430. Cellulose+fluoride and NC+fluoride showed antibacterial activity? It is very strange, to the best of my knowledge. I suppose that you should support your position. Probably, you can provide examples from literature other than your own article in Polymers in line 436? Those examples which are given in [32] are not related to the issue. Graphene oxide and CaO have completely different chemical nature than fluoride.
Thanks for valuable comments
Reference [32] was replaced by reference [77]
[77] Mitwalli H, Balhaddad AA, AlSahafi R, Oates TW, Melo MAS, Xu HHK, Weir MD. Novel CaF2 Nanocomposites with Antibacterial Function and Fluoride and Calcium Ion Release to Inhibit Oral Biofilm and Protect Teeth. J Funct Biomater. 2020 Aug 1;11(3):56
Minor comments:
Line 100. The symbols @ should be replaced with / or +.
Thanks for valuable comments
done
Line 121. NaOH, ethanol and HCl are not materials, they are chemicals. I was saying about discs and cards that they are materials, not chemicals.
Thanks for valuable comments
done
Line 128. Probably it should be a comma before ID-Gram-Positive...?
Thanks for valuable comments
No there isn’t
Line 146. Is there a space between Na and OH?
Thanks for valuable comments
No there isn’t
Line 250. It is better to replace the word peaks with absorption bands (or just bands) for FTIR in the text.
Thanks for valuable comments
done
Line 278. The answer for my previous question about EDS is better to insert into the manuscript: "So the cellulose appears as present C, and O, and Ag to form nanocomposites, the trace elements appear may be absorbed by algae and not dissolved during the nanocellulose preparations".
Thanks for valuable comments
done
Table 4, instead of writing 0.4 mg/mL in every cell, you would better write it in the name of the Table for not repeating it many times.
Thanks for valuable comments.
done
Line 429. Correct the commas/fullstops/spaces: No 48,8,40, and 25. 10.
Thanks for valuable comments.
Corrected
